



Atmospheric
Chemistry
and Physics

# Properties of ice cloud over Beijing from surface Ka-band radar observations during 2014–2017

**Juan Huo, Yufang Tian, Xue Wu, Congzheng Han, Bo Liu, Yongheng Bi, Shu Duan, and Daren Lyu**

Key Laboratory of Middle Atmosphere and Global Environment Observation, Chinese Academy of Sciences, Beijing, China

**Correspondence:** Juan Huo (huojuan@iap.ac.cn)

**Abstract.** The physical properties and radiative role of ice clouds remain one of the uncertainties in the Earth–atmosphere system. In this study, we present a detailed analysis of ice cloud properties based on 4 years of surface millimeter-wavelength radar measurements in Beijing, China, where the summer monsoon from the ocean and the winter monsoon from the continent prevail alternately, resulting in various ice clouds. More than 6300 ice cloud clusters were studied to quantify the properties of ice clouds, such as the height, optical depth and horizontal extent, which can serve as a reference for parameterization and characterization in global climate models. In addition, comparison between ice cloud clusters formed under the summer monsoon and the winter monsoon indicates the different formation and evolution mechanisms of cirrus clouds. Statistically, temperatures of more than 95 % of ice radar bins are below $-15\,°C$ and more than 80 % of ice clouds are above 7 km. The dependence of the radar reflectivity of ice particles on height and temperature was also observed in this study, indicating that the reflectivity of ice bins increases (decreases) as the temperature (height) increases. In addition, it is found that there is a strong linear relationship between the mean reflectivity and the ice cloud depth. Due to various synoptic circumstances, the ice clouds in summer are warmer, higher and thicker, with larger reflectivity than that in winter; in particular, the mean cloud-top height of ice clouds in summer is 2.2 km higher than that in winter. Our analysis indicates that in spring, in situ-origin cirrus clouds are more common than liquid-origin cirrus clouds, while in summer liquid-origin cirrus clouds are more frequent; in autumn and winter, most cirrus clouds are of in situ origin.

## 1 Introduction

The radiative role of ice clouds in the Earth–atmosphere system is known to be significant; however, uncertainties remain with respect to the net effects since ice clouds contain various types of nonspherical ice crystals (Yang et al., 2015). For example, ice clouds absorb the outgoing infrared radiation from Earth's surface and lower atmosphere while reflecting a portion of the incident sunlight back to outer space. When ice clouds are thin enough for the sun to be seen through them, the net impact on the planetary radiation balance is generally one of warming; thicker ice clouds reflect more sunlight and generally result in net cooling (Heymsfield et al., 2017; Kärcher, 2018; Kox et al., 2014). Cirrus clouds consist solely of ice crystals. Their occurrence frequency exhibits latitudinal variability ranging from 50 % in the equatorial regions of Africa to 7 % in the polar regions (Stubenrauch et al., 2006; Hahn and Warren, 2007; Sassen et al., 2008, 2009). Ice clouds cover over 50 % of the globe's surface (Hong and Liu, 2015). Dolinar et al. (2019) reported that single-layer ice clouds have a global occurrence frequency of about 18 %. Ice clouds are an important component of the planetary radiation budget in terms of magnitude; plus, they influence hydrological and climate sensitivities and affect surface climate (Runheng and Liou, 1985; Yang et al., 2015; Gultepe and Heymsfield, 2016; Lawson et al., 2019).

The physical and optical properties of ice clouds, such as ice crystal size, ice shape, particle concentration, cloud-top height (CTH) and optical depth, are heterogeneously and diversely distributed over the globe (Jensen et al., 1996; Mace et al., 2006; Yang and Fu, 2009; Adhikari et al., 2012; Cotton et al., 2013; Heymsfield et al., 2013; Luebke et al., 2016; Wolf et al., 2018; Ge et al., 2019). Recent studies show that cirrus clouds remain one of the largest sources of un-

certainty in global climate models (GCMs), due to the deficiencies in representing their observed spatial and temporal variability (Zelinka et al., 2012; Joos et al., 2014; Muhlbauer et al., 2014). According to an IPCC report (Boucher et al., 2013), "Especially for ice clouds, and for interactions between aerosols and clouds, our understanding of the basic microscale physics is not yet adequate, although it is improving". Understanding the microphysical and macrophysical properties of ice clouds, as well as their relationships with atmospheric states, such as temperature, wind velocity and relative humidity, is important for advancing our fundamental understanding of the formation and life cycles of ice cloud. It is also an essential step toward reducing the uncertainties in estimates of the climatic impact of cirrus clouds and improving the representation of ice clouds in GCMs. A better understanding of ice clouds is important for improving climate simulations and numerical weather predictions.

Millimeter-wavelength radar is a powerful method for observing the macroscopic and microphysical properties of vertical cloud profiles owing to its ability to penetrate the interior of clouds. Because of radar systems' short wavelengths, they are sensitive to small cloud droplets and ice crystals, meaning they detect all types of nonprecipitating clouds well (Kollias et al., 2007). Radar can perform long continuous observations, and the data have a high temporal resolution (i.e., detecting three profiles per second with the vertically pointing mode), which is more advantageous than aircraft in understanding the characteristics of daily changes and the formation and development of clouds. Regular calibration of radar instruments can ensure the stability of data and support long-term data for cloud climatology research. This study used long-term, continuous, surface Ka-band radar data to study and understand the microphysical and macrophysical properties of ice clouds over Beijing, China, in the northern midlatitude region. Beijing (39.96° N, 116.37° E) is in the subtropical monsoon zone with a typical continental monsoon climate. Winds from the southeast ocean prevail in summer, while winds from the northwest continent dominate in winter, resulting in hot and rainy summers but cold and dry winters. The formation, evolution and life cycle of ice clouds present regional and distinctive traits, which are created by the regional climate and, to a certain extent, the global climate too. This paper presents the features of ice clouds over midlatitude monsoon regions through detailed analysis based on long-term radar data and serves as a reference for cloud parameterization in GCMs.

Section 2 of this paper briefly introduces the Ka-band radar data, the identification method for ice clouds and other auxiliary datasets. Section 3 describes the macrophysical properties of ice clouds. Details of the microphysical properties of ice clouds are presented in Sect. 4. In Sect. 5, the formation types of cirrus clouds in four seasons are investigated. Conclusions are given in Sect. 6.

## 2 Data and method

### 2.1 Ka-band radar

The ice clouds analyzed in this study are from observations of a Ka-band polarization Doppler radar (KPDR) situated at the Institute of Atmospheric Physics (IAP; 39.967° N, 116.367° E), Beijing, China. The KPDR was set up in 2010 and works at a frequency of 35.075 GHz (wavelength of 8.55 mm; Huo et al., 2019), measuring the equivalent reflectivity factor ($Z_e$, hereinafter simply "reflectivity"; units $mm^6 m^{-3}$; $dBZ = 10\log(Z_e)$), Doppler velocity, spectral width and linear depolarization ratio of cloud. It is equipped with a magnetron-type transmitter with a minimum sensitivity of $-45$ dBZ for cloud determination. For comparison, the 94 GHz cloud profiling radar (CPR) on CloudSat has a sensitivity of approximately $-30$ dBZ. Calculations or measurements of radar reflectivity in previous studies reveal that the reflectivity of ice clouds over midlatitude regions are mostly larger than $-30$ dBZ (Deng et al., 2010; Pokharel and Vali, 2011; Matrosov and Heymsfield, 2017). Therefore, the KPDR is capable of detecting most ice clouds over Beijing. However, the Ka-band radar is more sensitive to larger particles in a cloud target since the reflectivity is proportional to the $D^6$ ($D$ is particle size). For the CPR, thin ice clouds with ice water content (IWC) lower than approximately $0.4 \, mg \, m^{-3}$ are invisible (Wu et al., 2009). It is possible that the KPDR misses some thin ice clouds when they consist of small ice crystals (i.e., $D$ less than 20 μm) or the IWC is smaller than $0.4 \, mg \, m^{-3}$. The pulse width of the KPDR is 0.2 μs, and the beamwidth is 0.4°. Its repetition frequency is 3.5 kHz, and its vertical resolution is 30 m. The KPDR has operated daily since 2012, mostly in the vertically pointing mode. During special events – for example, short-term collaborative observations with other instruments – the scanning mode changes to the plane position indicator or radar height indicator mode. In 2013 and 2018, the KPDR was nonoperational during almost the whole of the summer period. The radar data used in this paper were observed from 1 January 2014 to 31 September 2017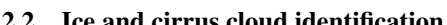. During these 4 years, the valid operational time of the radar in the vertically pointing mode occupied more than 80 % of the total time. Namely, there are more than 28 000 h of radar measurements in the vertically pointing mode during the period 2014–2017.

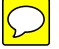

### 2.2 Ice and cirrus cloud identification

Ice clouds are composed of various types of ice crystals and are usually thin. Cirrus clouds also consist of ice crystals. Definitions of cirrus clouds in previous publications provide us with references for ice cloud identification with the KPDR. For example, in the cloud classification algorithm developed for the CPR on the CloudSat satellite, the average temperature at the largest $Z_e$, the average height of the max-

imum $Z_e$, the cloud-base height (CBH), etc. are combined to determine cirrus cloud (Wang and Sassen, 2001b). Sassen et al. (2008) classified cloud layers as cirrus via defining two criteria: namely, the visible optical depth should be less than 3.0 and the cloud-top temperature should be lower than $-40\,°\text{C}$, categorizing cirrus clouds via cloud physical and optical parameters. Deng et al. (2010) identified cirrus layers by cloud-top and cloud-base temperature ($T_{\text{top}} < -40\,°\text{C}$ and $T_{\text{base}} < -10\,°\text{C}$). In some recent studies, cirrus clouds are defined as ice clouds with temperatures $< -38°$ (Krämer et al., 2016; Luebke et al., 2016; Heymsfield et al., 2017). Ge et al. (2019) used two temperature criteria to identify cirrus cloud: the temperature of the cloud top should be less than $-30\,°\text{C}$, and the temperature at the maximum $Z_e$ layer and at the cloud base should be less than $0\,°\text{C}$. The KPDR has a cloud clustering and classification algorithm, a detailed description of which has been presented by Huo et al. (2019). Here, we briefly describe it as follows. Firstly, the KPDR cloud profiles are grouped as clusters based on a combination of a time–height clustering method and a $k$-means clustering method. After each cloud cluster is determined, a fuzzy-logic method is applied using multiple cloud properties, such as CBH, cloud depth (CD) and radar reflectivity, to classify the cloud cluster into nine types: Cs, Cc, Ac, As, St, Sc, Ns, Cu and Cb clouds. According to the definitions and identification approaches in previous studies, we use two criteria to identify ice clouds from KPDR data after the clustering and classification algorithm is performed. Namely, a cloud cluster for which the mean cloud-top temperature is less than $-40\,°\text{C}$ and the maximum cloud-base temperature is less than $-10\,°\text{C}$ is determined as ice cloud. Ice cloud with a cloud-base temperature below $-38\,°\text{C}$ is regarded as cirrus cloud in this paper. It should be noted that supercooled water might exist in ice clouds with a temperature above $-38\,°\text{C}$, and thus what the radar measures should indicate different physical properties from that of ice particles. In this paper, the supercooled water is not distinguished, and its proportion and properties will be investigated in the future.

## 2.3 Other datasets

This study also used some other datasets to complement the investigation of the properties of ice and cirrus cloud, such as the temperature profile, water vapor, wind velocity and cloud optical thickness. The research datasets of cloud optical thickness (produced from Himawari-8) used in this paper were supplied by the P-Tree System of the Japan Aerospace Exploration Agency (https://www.eorc.jaxa.jp/ptree/index.html, last access: 6 January 2020). Other meteorological reanalysis data employed were from the European Centre for Medium-Range Weather Forecasts (ECMWF) ERA5 datasets (https://www.ecmwf.int/en/forecasts/datasets/reanalysis-datasets/era5, last access: 6 January 2020).

## 3 Macrophysical properties

### 3.1 Ice cloud samples under the summer and winter monsoon

Ice clouds can be vertically and horizontally extensive, with their various appearances dependent on the diverse range of associated atmospheric movements and processes. The KPDR is located in the north of the North China Plain, where to the west and north there are mountains and to the south and east is the Bohai Sea. In the region's hot summers, the monsoon from the sea brings large quantities of water vapor, whereas the dry and cold monsoon from the northern continent dominates this region in winter. These different monsoon types support various atmospheric conditions, such as increasing relative humidity, cooling and updrafts, required for the formation of ice clouds, ultimately resulting in distinct cirrus distributions. Figure 1 presents a typical example of an ice cloud distribution collected by the KPDR in 1 month of winter (January 2016) and 1 month of summer (August 2015).

There are more ice clouds in August than in January, and the mean radar reflectivity of ice cloud in August is higher than that in January. Ice clouds in August also show larger vertical dimensions than in January. The temperature and amount of water vapor are two key parameters in the formation of clouds, especially in plain areas where orographic uplift is negligible. The strong contrast in the climatic circumstances between a month in summer and a month in winter generates a diverse range of ice clouds (Fig. 1c). Thus, to better understand the physical or optical properties of ice clouds, statistical analyses were carried out in this study for different seasons. Such comparisons of the ice clouds among the four seasons benefit our understanding of the dominant formation origins of ice clouds when a region is governed alternately by different monsoon types. In this study, 4 years of radar observations presented more than 6300 ice cloud clusters for our analysis.

### 3.2 Monthly and hourly occurrence frequency

Radar data collected in vertically viewing mode were used to calculate the occurrence frequency of all clouds ($O_{\text{all}}$), which is the ratio of cloudy profiles to all profiles in a certain time range (i.e., an hour or a month), as well as the occurrence frequency of ice clouds ($O_{\text{ic}}$), which is the ratio of profiles determined as ice clouds to all radar profiles:

$$O_{\text{all}} = N_{\text{all}}/N_{\text{r}}, \tag{1}$$

$$O_{\text{ic}} = N_{\text{ic}}/N_{\text{r}}, \tag{2}$$

where $N_{\text{all}}$ is the number of cloudy profiles, $N_{\text{r}}$ is the number of all radar profiles and $N_{\text{ic}}$ is the number of ice cloud profiles. Figure 2 shows the monthly occurrence frequency of all clouds and ice clouds in 4 years. In addition, the occurrence frequency of cirrus clouds is also presented for con-

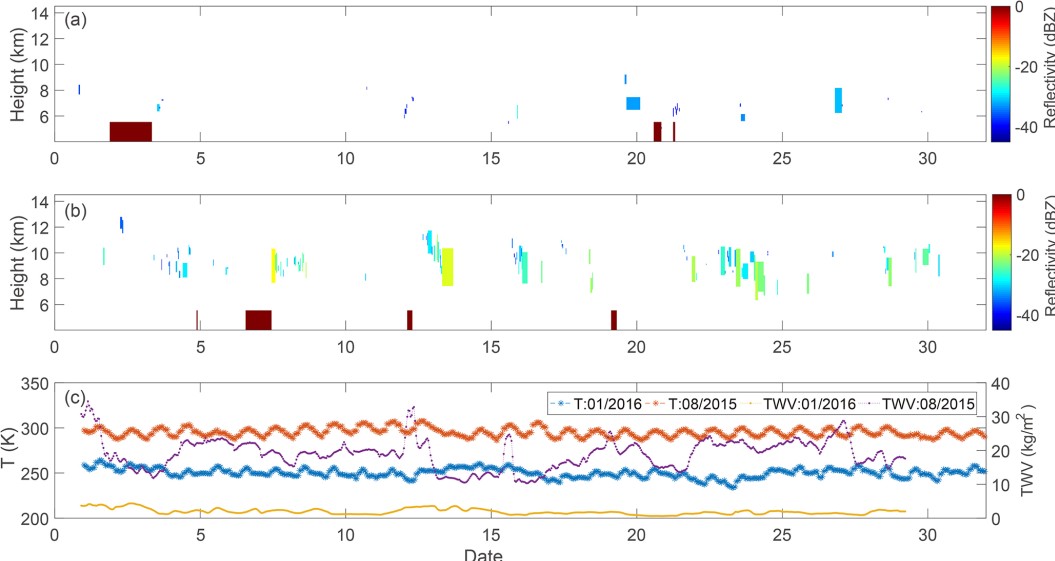

**Figure 1.** Ice clouds occurring in **(a)** January 2016 (winter) and **(b)** August 2015 (summer). The mean cloud-top height, mean base height and lifetime of each ice cluster forms an ice cloud "rectangle". Its mean radar reflectivity is illustrated with different colors. Dark red rectangles on the horizontal axis indicate periods without vertically pointing radar measurements. The surface temperature ($T$; left-hand $y$ axis) and total water vapor (TWV; right-hand $y$ axis) in the 2 months are presented in panel **(c)**.

trast. September has the maximum $O_{all}$ among all months, and summer/winter has the maximum/minimum $O_{all}$ among the four seasons. Relative to $O_{all}$, $O_{ic}$ decreases to 33 %– 50 %, and in winter $O_{ic}$ is about 33 % of $O_{all}$. The average $O_{ic}$ in April and June is about 20 %, whereas in winter (December–February) it is no more than 10 %. The average $O_{ic}$ in the 4 years is 14 %, which is lower than the ice cloud coverage of 24 % reported by Hahn and Warren (2007) based on satellite measurements over North China. This might be associated with the observation location and the field of view (FOV) of the KPDR. Large quantities of water vapor over the sea areas and orographic-lift movements over mountain areas provide advantageous conditions for the formation of clouds, meaning more clouds occur over these areas relative to plain areas. Therefore, the occurrence frequency calculated from the KPDR data with a small FOV is lower than the cloud coverage calculated from data with a broad FOV. For cirrus clouds, the largest occurrence frequency (4 %) occurs in April. Spring, but not summer, has the most cirrus clouds.

The KPDR operates continuously and thus allows the diurnal variation in $O_{ic}$ to be studied, which illustrates the potential relationship with local thermal convection caused by solar heating. As shown in Fig. 2a, the three highest $O_{ic}$ values in spring, summer, autumn and winter occur at 20:00/22:00/19:00, 21:00/23:00/22:00, 00:00/22:00/21:00 and 14:00/13:00/17:00 LT (UTC+8), respectively. The hourly variations in $O_{ic}$ in the four seasons are different; in spring, summer and autumn, larger $O_{ic}$ values appear at night, whereas larger $O_{ic}$ values in winter appear during the daytime. The diurnal variation in $O_{ic}$ seems

to be insensitive to solar heating, which drives the development of regional thermal convection. Here, the presence of ice clouds over the KPDR is not closely related to local air-updraft activities, indicating that these ice clouds may mostly not be generated locally by thermal convections. It is interesting that $O_{ic}$ decreases from 00:00 to 02:00 LT and then increases after that in the four seasons. Is there a decay process in ice clouds during this period? Is the decrease caused by wind, vertical movement or turbulence? Further analysis is required in the future to answer these questions.

### 3.3 Height, depth and extent

The top height of ice cloud, especially cirrus cloud, indicates the highest condensation level in the troposphere, above which clouds cannot form because of the nonconducive condensation conditions. The base height of ice clouds indicates the lowest level required for ice formation. In this study, the CTH and CBH were calculated for each ice cloud cluster; specifically, the CTH and CBH are the mean values of all cloudy profiles in an ice cloud cluster. The distributions of the mean CTH and CBH of all ice clouds in the four seasons are presented in Fig. 3, and Table 1 presents the quantified statistical results.

It is shown that the CTH of ice clouds varies in the range of 5.5–12.9 km (Fig. 3a). The difference between the maximum and the minimum is about 6 km in each season, indicating the ranges of the condensation level and various formation mechanisms of ice clouds. Besides, differences in the CTH between the four seasons are also apparent. Both the maximum (12.9 km) and the highest mean (9.27 km) CTH

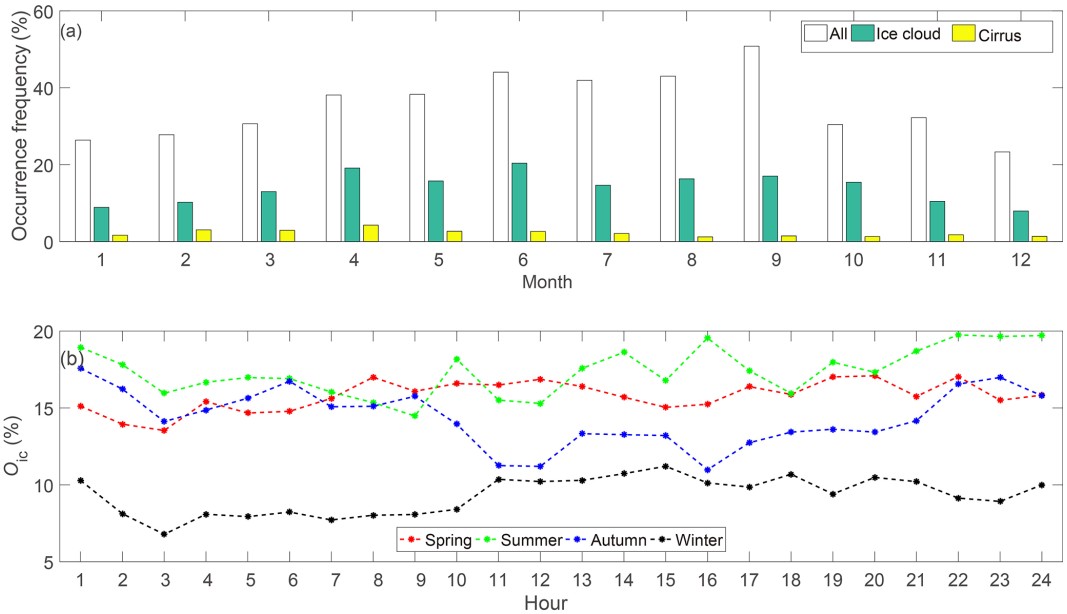

**Figure 2.** Monthly occurrence frequencies of all clouds ($O_{all}$), ice clouds ($O_{ic}$) and cirrus clouds (**a**), along with the diurnal $O_{ic}$ in the four seasons (**b**).

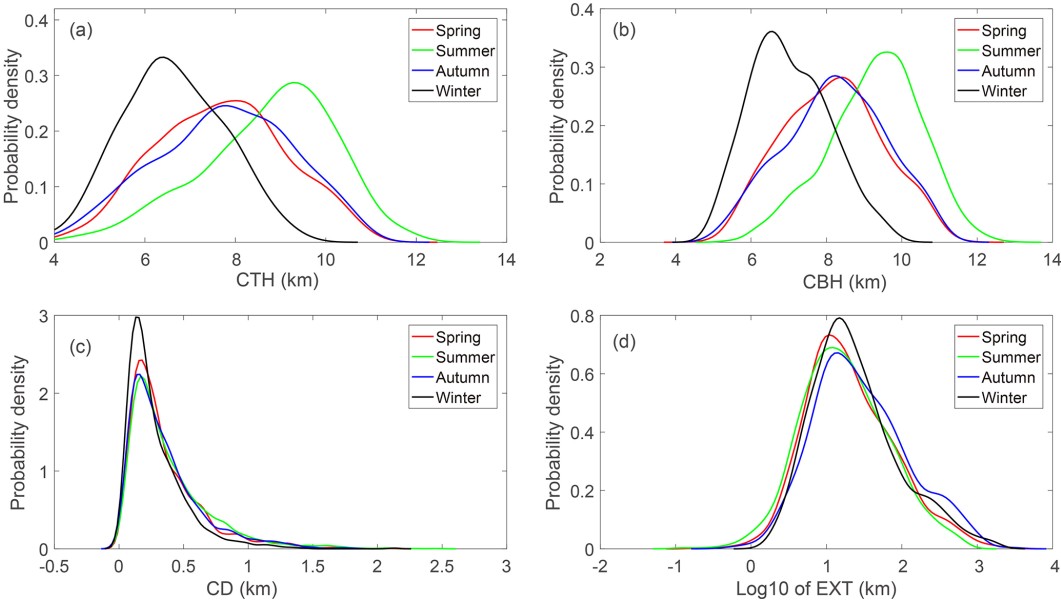

**Figure 3.** Distribution of cloud-top height (**a**), cloud-base height (**b**), cloud depth (**c**) and horizontal extent (EXT; **d**) in the four seasons. In panel (**d**), EXT is shown as log10 values.

are found in summer, whereas winter has the minimum CTH (9.94 km) and lowest mean CTH (7.02 km). In summer, 85 % of ice clouds have a CTH greater than 8 km and 29 % are greater than 10 km. In winter, 71 % of ice clouds have a CTH larger than 8 km and those with a CTH higher than 6 km account for 97 %. The mean CTH in summer is 2.2 km higher than that in winter, which means the average condensation level in summer is also 2.2 km higher. Spring and autumn are

two transition seasons and their CTHs are 8.16 and 8.23 km, respectively, which are between those of summer and winter.

Figure 3b shows that the CBH changes within a range of 5.3–12.4 km, and the minimum CBHs in the four seasons are close to each other, ranging between 5.3 and 5.7 km. However, the mean CBH in summer is the highest (8.7 km) among the four seasons, while the lowest (6.6 km) is in winter. The difference in CBH between summer and winter is 2.1 km.

**Table 1.** Statistical results for the cloud-top height (CTH), cloud-base height (CBH), cloud depth (CD), horizontal extent (EXT) and cloud optical depth (COD) in the four seasons. The "trimmean" is the 10 % trimmed mean of portion clusters, excluding 10 % of clusters with the highest and lowest values (unit: km).

| Season | Parameters | Mean | Median | Trimmean | Maximum | Minimum |
|--------|-----------|------|--------|----------|---------|---------|
| Spring | CTH | 8.16 | 8.17 | 8.15 | 11.74 | 5.73 |
|        | CBH | 7.68 | 7.68 | 7.68 | 11.43 | 5.43 |
|        | CD | 0.35 | 0.27 | 0.32 | 2.1 | 0.06 |
|        | EXT | 61.5 | 17.34 | 35.6 | 2824.9 | 0.18 |
|        | COD | 4.27 | 3.22 | 3.81 | – | 0.06 |
| Summer | CTH | 9.27 | 9.38 | 9.30 | 12.86 | 6.10 |
|        | CBH | 8.73 | 8.97 | 8.78 | 12.42 | 5.64 |
|        | CD | 0.39 | 0.30 | 0.35 | 2.45 | 0.06 |
|        | EXT | 43.0 | 16.1 | 29.6 | 725.1 | 0.13 |
|        | COD | 6.07 | 4.28 | 5.64 | – | 0.1 |
| Autumn | CTH | 8.23 | 8.27 | 8.24 | 11.25 | 5.69 |
|        | CBH | 7.74 | 7.80 | 7.77 | 11.07 | 5.31 |
|        | CD | 0.35 | 0.28 | 0.33 | 1.82 | 0.06 |
|        | EXT | 86.10 | 23.5 | 55.17 | 2863.8 | 0.47 |
|        | COD | 4.62 | 3.05 | 4.01 | – | 0.01 |
| Winter | CTH | 7.02 | 6.90 | 7.00 | 9.94 | 5.50 |
|        | CBH | 6.63 | 6.57 | 6.63 | 9.75 | 5.30 |
|        | CD | 0.28 | 0.21 | 0.26 | 2.13 | 0.04 |
|        | EXT | 72.7 | 19.3 | 41.4 | 1695.2 | 1.50 |
|        | COD | 4.52 | 2.80 | 4.10 | – | 0.21 |

The mean CBHs in spring and autumn are both 7.7 km. In summer, the percentage of ice clouds with a CBH larger than 8 km is 72 %, while it is only 65 % in winter. In summer and winter, 95 % of ice clouds have a CBH greater than 6 km.

It is shown that the mean CDs of ice clouds in the four seasons are close, with the depths of most clusters being less than 1 km (Fig. 3c). Statistically, in the four seasons, 68 % of clusters have a CD of less than 0.5 km, 90 % have a CD of less than 1 km and 96 % have a CD of less than 1.5 km. It should be noted that the CTH, CBH and CD here are the mean values of an ice cloud cluster. It is therefore possible that there are some instances of CTH, CBH and CD that are greater than their corresponding mean values.

The horizontal extent (EXT) of ice clouds indicates the clouds' lifetimes and their formation mechanism type. For the KPDR, the EXT of an ice cloud cluster is computed as follows:

$$\text{EXT} = V_{\text{hw}} \times T_{\text{ci}}, \tag{3}$$

where $V_{\text{hw}}$ is the mean velocity of horizontal wind calculated from the ECMWF-ERA5 dataset and $T_{\text{ci}}$ is the continuous time during which an ice cluster moves over the KPDR. It is found that the maximum EXT of ice clouds reaches 2800 km, in April 2017, and the maximum $T_{\text{ci}}$ is 21 h, in March 2016. The EXT ranges through orders of magnitude from low values of less than 0.1 km to over 2800 km. Summer has the minimum mean, median and trimmed mean

EXT, while ice clouds in autumn have the maximum mean, median and trimmed mean EXT. Statistically, about 75 % of ice clouds have an EXT of less than 50 km and 87 % have an EXT of less than 100 km. The statistically quantified structural properties of ice clouds in the four seasons are presented in Table 1.

### 3.4 Optical depth

Cloud optical depth (COD) is relatively independent of wavelength throughout the visible spectrum. In the visible portion of the spectrum, the COD is almost entirely due to scattering by droplets or crystals of clouds (American Meteorological Society, 2019). Therefore, the CODs of ice clouds depend directly on the CD, the IWC and the size distribution of the ice crystals, indicating a cooling effect or warming effect in the energy budget.

The Advanced Himawari Imager (AHI), on board the geostationary meteorological Himawari-8 satellite operated by the Japan Meteorological Agency, observes the Beijing area every 10 min and began releasing COD and cloud-type products in July 2015 with a spatial resolution of 5 km. The CODs are retrieved by using nonabsorbing visible wavelengths (i.e., 0.51 or 0.64 μm) and water-absorbing, near-infrared wavelengths (i.e., 1.6 or 2.3 μm; Nakajima and Nakajma, 1995; Kawamoto et al., 2001). Quantified uncertainties in the AHI CODs have not been reported, so we use them here directly. The data nearest to the KPDR that the AHI determines as

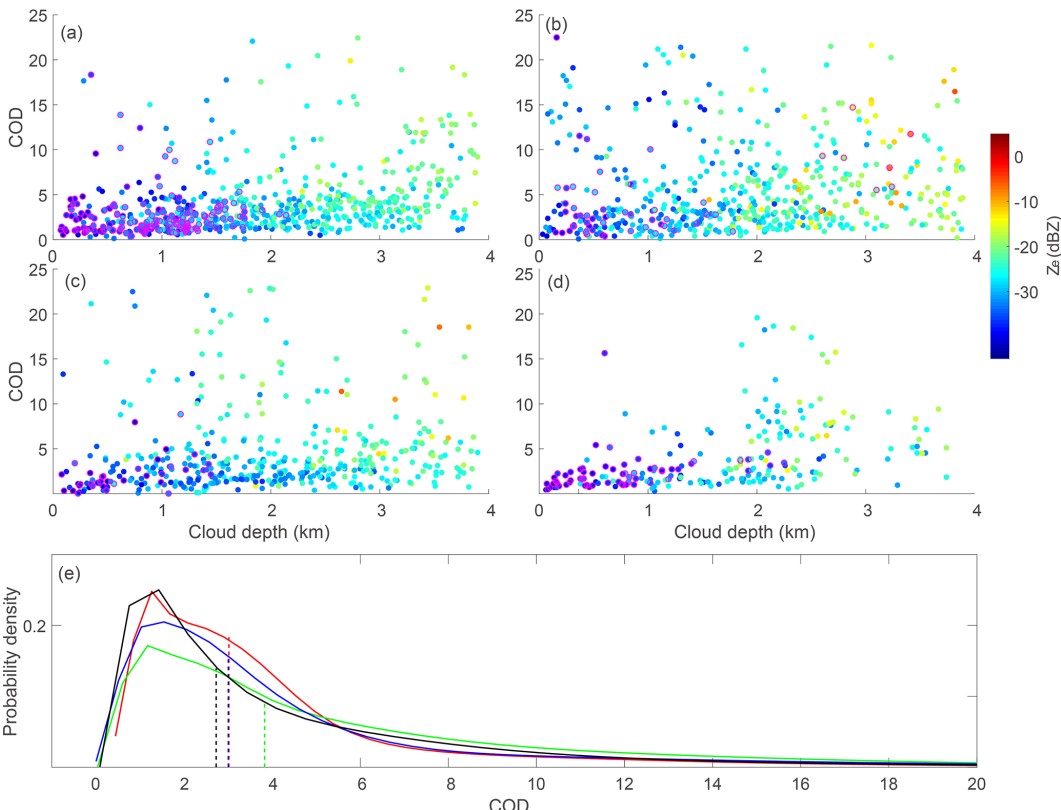

**Figure 4.** The cloud optical depth (COD) of ice clouds in terms of the cloud depth in spring **(a)**, summer **(b)**, autumn **(c)**, and winter **(d)**. Colors indicate the mean reflectivity of those radar profiles within 10 min of the AHI (Advanced Himawari Imager) observation time. Cases with cloud-base temperatures lower than −38 °C are illustrated with purple edges. Panel **(e)** presents the probability density distribution of the COD in the four seasons.

cirrus cloud and the KPDR determines as ice cloud are selected, and their CODs are investigated. Those collocated CODs (collected from the year 2016 to 2017) combined with the mean CDs and mean reflectivity, which are calculated from all cloudy KPDR bins observed within 10 min of the AHI observing time, are presented in Fig. 4.

In the four seasons, CODs show an increasing tendency with increasing CD. The mean reflectivity shows a similar tendency, meaning thicker ice clouds generally contain larger particles and a greater number density of ice particles. The probability density distributions of CODs in the four seasons show a higher probability occurring at lower CODs. The mean COD in spring, summer, autumn and winter is 4.27, 6.07, 4.62 and 4.52, respectively. The proportions of CODs lower than 3 in spring, summer, autumn and winter are 46 %, 36 %, 49 % and 52 %, respectively. The proportions of CODs lower than 10 in the four seasons are 91 %, 79 %, 87 % and 90 %, respectively. In Fig. 4, KPDR cirrus clouds are shown with purple circles. The proportions of CODs for cirrus clouds lower than 3 in spring, summer, autumn and winter are 70 %, 55 %, 77 % and 79 %, respectively. The proportions of CODs lower than 6 in the four seasons are 93 %, 77 %, 94 % and 98 %, respectively. As shown in previous

studies, the primary cirrus optical depth is below 3 (Kienast-Sjögren et al., 2016; Sassen et al., 2008). Here, our analysis shows larger CODs, indicating possible mixed-phase clouds. Also, it might be related to the uncertainty in CODs, as the uncertainties in AHI CODs and the different FOV between the KPDR and the AHI may cause the employed CODs to differ from the real CODs. The results here present statistical features of the CODs of ice clouds over Beijing based on a COD dataset of limited accessibility. For cirrus clouds, lidar should provide accurate COD measurements.

## 4   Microphysical properties

The most important microphysical quantities of ice clouds are the ice particle size distribution, the IWC and the clouds' shapes (Heymsfield et al., 2017). It is known that the radar equivalent (or effective) reflectivity factor can be expressed as

$$Z_e = \frac{\lambda^4}{\pi^5} \left| \frac{m^2 - 1}{m^2 + 2} \right|^{-2}$$
$$\iiint \sigma(D, \theta, \Phi) N(D, \theta, \Phi) \, dD d\theta d\Phi, \qquad (4)$$

where $\sigma(D, \theta, \varnothing)$ is the backscattering cross section with maximum dimension $D$ and an axial direction $(\theta, \varnothing)$ with respect to the radar beam, $N(D, \theta, \varnothing)$ is the number density, $\lambda$ is the wavelength, and $m$ is the complex index of refraction of the scattering target. To date, numerous empirical relationships between $Z_e$ and cloud properties $(P)$ – e.g., IWC, snow precipitation rate – have been developed, usually in the power-law form of

$$Z_e = A P^B, \qquad (5)$$

where $A$ is the prefactor coefficient and $B$ is the exponent derived in terms of calculated or measured datasets (Liu and Illingworth, 2000; Wang and Sassen, 2001a; Heymsfield et al., 2008; Austin et al., 2009; Delanoë and Hogan, 2010; Deng et al., 2015; Matrosov and Heymsfield, 2017; Heymsfield et al., 2018). Delanoë and Hogan (2008, 2010) proposed a different method using a forward model to retrieve the IWC and the effective radius by combination with the COD. Also, the basic principles of this method are applied in the CloudSat–CALIPSO cloud microphysical retrieval algorithm. However, the use of empirical relations such as in Eq. (5) is still common in many practical measurements, and the correspondence between the IWC and $Z_e$ is related to the particle size distribution (the gamma distribution is mostly used for ice clouds).

For the KPDR, the development of the IWC and particle size retrieval algorithm is in progress but has not been tested completely. In this paper, we use the measured radar reflectivity factor $Z_e$ directly, not the retrieved microphysical quantities, to study and characterize the microphysical properties of ice clouds. It can be found from Eq. (2) that reflectivity increases when $\sigma$ and $N$ increase; in other words, a larger reflectivity normally indicates a larger $D$, $N$ and IWC.

### 4.1 Reflectivity and height dependence

The KPDR detects clouds at a 30 m vertical resolution. All ice radar bins collected from 2014 to 2017 were counted according to their reflectivity and height, and the relative frequencies (counts; calculated at 0.25 dBZ and 30 m intervals within 15 km) are shown separately in Fig. 5. As presented, ice clouds exist below the height of 13 km. In summer, the reflectivity mostly varies between $-35$ and $-10$ dBZ, while most of the reflectivity falls within the range of $-40$ to $-20$ dBZ in winter. In spring and autumn, the reflectivity primarily ranges between $-35$ and $-15$ dBZ. The range of variation in reflectivity in summer is the biggest among the four seasons, while it is the smallest in winter. Statistically, at the same height where ice clouds exist in the four seasons, the mean reflectivity of winter is 5 dBZ less than that of spring or autumn and it is 10 dBZ less than that of summer. In the four seasons, the mean reflectivity declines as the height increases below 11 km, with a similar slope. At the height above 11 km, the relationship between reflectivity and the height varies greatly among the four seasons, which

might be due to the small sample counts. It can also be seen that the ice bins in summer are located at higher heights than in winter.

### 4.2 Temperature dependence

Temperature plays a key role in the formation, evolution and lifetime of ice clouds. Activation of liquid water drops does not happen below $-38\,°C$ because the relative humidity where the ice forms is below water saturation. At temperatures higher than $-38\,°C$, primary ice clouds form only when aided by ice-nucleating particles (Kanji et al., 2017). The summer monsoon and winter monsoon in Beijing support distinct temperatures, water vapor, etc., i.e., the conditions necessary for the formation of ice clouds, resulting in distributions of reflectivity with different features corresponding to temperature. The frequencies (counts; calculated at 0.25 dBZ and 1 °C intervals) are shown separately in Fig. 6.

In spring, summer and autumn, ice clouds occur mostly at temperatures within the range of $-10$ to $-55\,°C$, relative to which ice clouds in winter occur at lower temperatures. Statistically, the percent of ice bins with temperatures less than $-15\,°C$ is 99 %, 95 %, 95 % and 99 % in spring, summer, autumn and winter, respectively; the percent of ice bins with temperatures less than $-25\,°C$ is 85 %, 71 %, 72 % and 92 % in spring, summer, autumn and winter, respectively; and the percent of ice bins with temperatures less than $-35\,°C$ is 52 %, 36 %, 35 % and 60 % in spring, autumn and winter, respectively. The reflectivity shows a dependence on the temperature, increasing when temperature increases. Statistically, the mean temperature of ice clouds in winter is lower than that in other seasons, even though these ice clouds appear at lower heights. As the temperature decreases, the difference in reflectivity between winter and summer declines. At the same temperature, the mean reflectivity in summer is higher than that in winter. The slopes among the four seasons are close to each other when temperature is above $-50\,°C$, demonstrating a determinative effect of the temperature on the cloud particle properties. The slopes in the four seasons disperse at temperatures lower than $-50\,°C$, which might be because the small sample counts influence the representativeness of statistical results.

### 4.3 Depth dependence

Based on all the ice clusters in the 4 years, we calculated the mean reflectivity and the mean depth of each cluster (Fig. 7), and it was interesting to find that there is a strong linear relationship between the mean reflectivity and the CD. Specifically, the mean reflectivity increases as the CD increases. The linear equation shown in Fig. 7 represents a method that can be used to estimate the mean reflectivity (or CD) if the CD (or reflectivity) is known, which should be useful for cloud parameterization in GCMs.

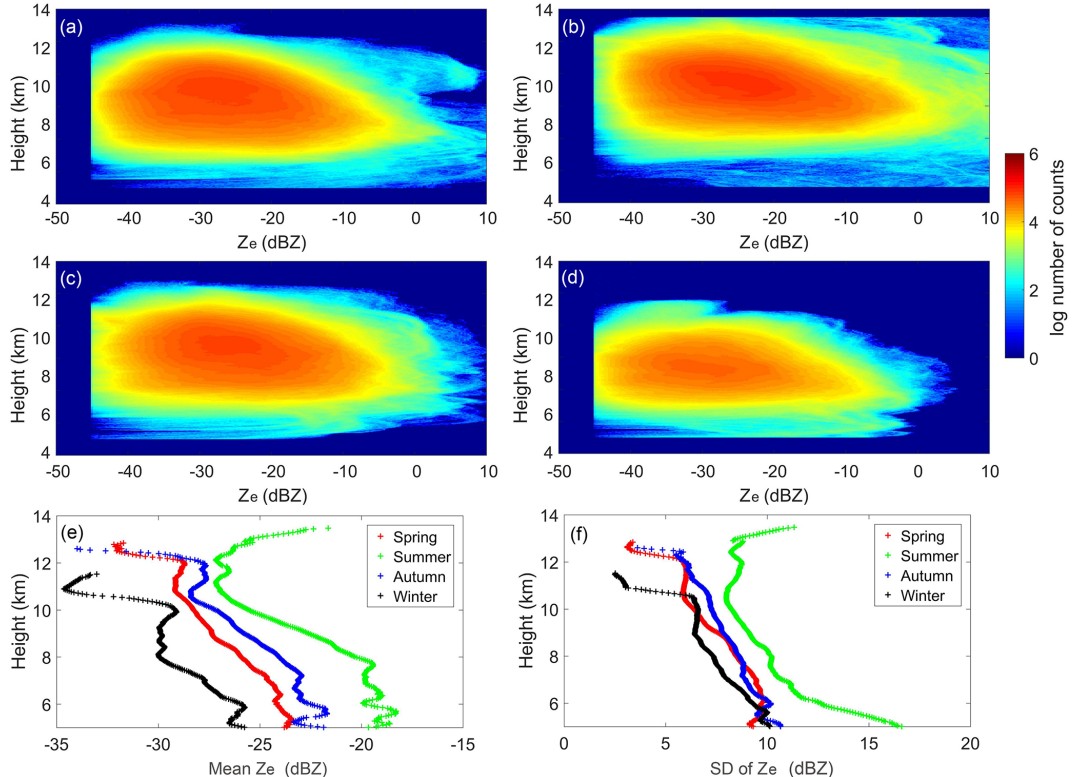

**Figure 5.** The frequency of reflectivity versus height in spring (**a**), summer (**b**), autumn (**c**) and winter (**d**). Colors are the log number of the counts (calculated at 0.25 dBZ and 30 m intervals). The mean reflectivity calculated at various heights and the corresponding standard deviation (SD) are presented in panels (**e**) and (**f**), respectively.

## 5 Origination type of cirrus clouds

Krämer et al. (2016) and Luebke et al. (2016) classified two types of cirrus cloud according to their formation mechanism; namely, in situ- and liquid-origin cirrus cloud. The in situ-origin cirrus type forms directly as cirrus clouds, while the liquid-origin type originates from mixed-phase clouds that are completely frozen until they are lifted to temperatures of $< -38\,^{\circ}$C. Krämer et al. (2016) and Luebke et al. (2016) reported that the in situ-origin cirrus clouds are mostly thin, with a lower IWC, while liquid-origin cirrus clouds tend to be thicker with a higher IWC. Also, liquid-origin cirrus clouds tend to have larger ice crystals than in situ-origin cirrus clouds. Various prefactor coefficients dependent on temperature have been derived and applied in the $Z_e$–IWC power-law relationship (i.e., Eq. 5) since the distribution of reflectivity has a dependence on temperature, just as shown above in Sect. 4.2 (Hogan et al., 2006; Heymsfield et al., 2013, 2018; Matrosov and Heymsfield, 2017). Therefore, the reflectivity of in situ-origin cirrus clouds should generally be less than that of liquid-origin cirrus clouds.

In this section, based on the frequency statistics in Sect. 4.2, we calculated the distribution of reflectivity (similar to the probability density function, PDF) at several temperatures to investigate the origin type of cirrus clouds in Bei-

jing. Figures 8–11 show the normalized frequency of reflectivity at central temperatures of $-65$, $-60$, $-55$, $-50$, $-45$ and $-40\,^{\circ}$C within $\pm 1\,^{\circ}$C in spring, summer, autumn and winter, respectively. The maximum counts, $n$, used to normalize the frequency is also presented in the figures.

Cirrus clouds present diverse reflectivity frequency distributions in terms of temperature in the four seasons. There is no cirrus cloud detected in summer and autumn (see Figs. 9a and 10a) at or below $-65\,^{\circ}$C. The number of cirrus cloud bins at $-65\,^{\circ}$C in spring and autumn is very small when compared to other temperatures above $-50\,^{\circ}$C. This might be because the frequency of atmospheric temperature $< -65\,^{\circ}$C within the troposphere over Beijing is small, so cirrus clouds at and below this temperature are few. Cirrus clouds also occur little in summer and autumn at $-60\,^{\circ}$C, due to the higher average temperature than in the other two seasons. Above $-55\,^{\circ}$C, the peak frequency center in winter is located at a smaller reflectivity value than that in summer, indicating smaller particles and a smaller number density than in summer. In the four seasons, the $n$ value at $-45\,^{\circ}$C is the biggest among all temperatures, indicating that cirrus cloud appears more frequently at $-45\,^{\circ}$C than at other temperatures. The reason is that at these altitudes both in situ-origin and liquid-origin cirrus clouds appear, whereas at colder temperatures only in situ-origin cirrus clouds exist (Krämer et al., 2020).

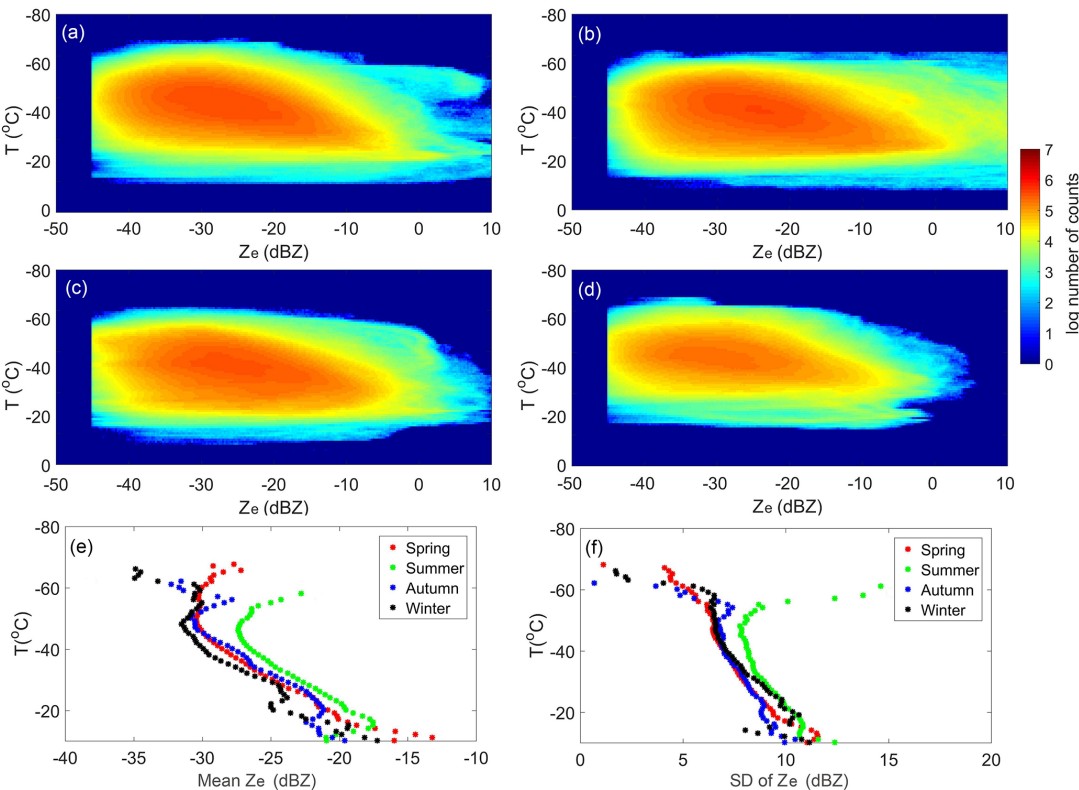

**Figure 6.** As in Fig. 5 but for temperature, and the colors are the log number of the counts (calculated at 0.25 dBZ and 1 °C intervals).

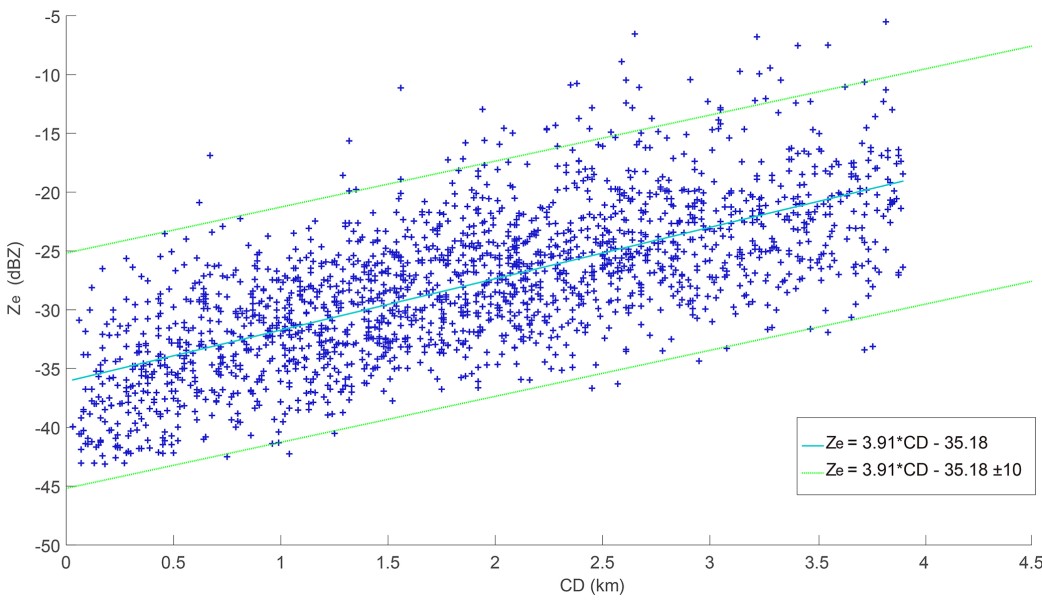

**Figure 7.** The mean reflectivity of ice clouds as a function of cloud depth (CD).

Spring has the biggest $n$ value at each temperature, indicating that cirrus clouds in spring are the most frequent, which has also been shown in Fig. 2.

A bimodal PDF is found at some temperatures – for example, at −60 and −65 °C in spring and at −45 °C in au-

tumn. However, most PDFs show a unimodal feature. One possibility is that only one origin type exists in Beijing. Another possibility is that the difference between the two origin types is not clearly distinguished. It might be related to the measurement specialties of radar since the $Z_e$ indicates the

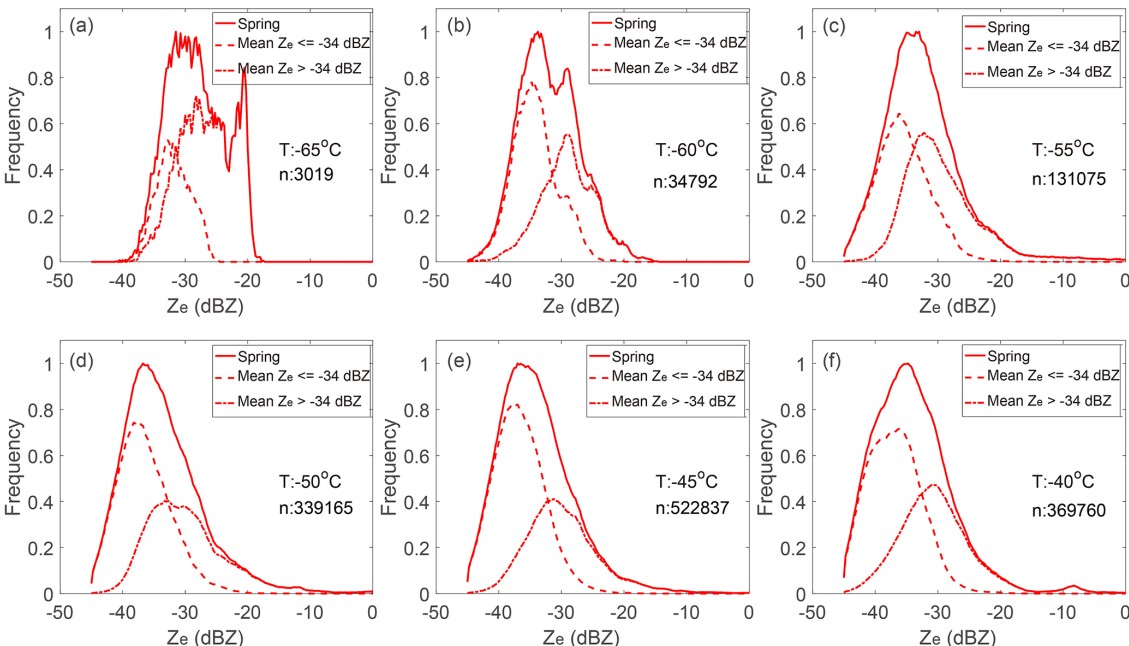

**Figure 8.** Normalized frequency of the reflectivity at different temperatures ($T$) in spring. The solid line is the frequency calculated from all cirrus clouds. The dashed line is the frequency calculated from cirrus clouds with a mean $Z_e < -34$ dBZ. The dash-dotted line is the frequency calculated from cirrus clouds with a mean $Z_e$ above $-34$ dBZ. The dashed line corresponds to the in situ-origin type, and the dotted line corresponds to the liquid-origin type. $n$ is the maximum number used to normalize the frequency.

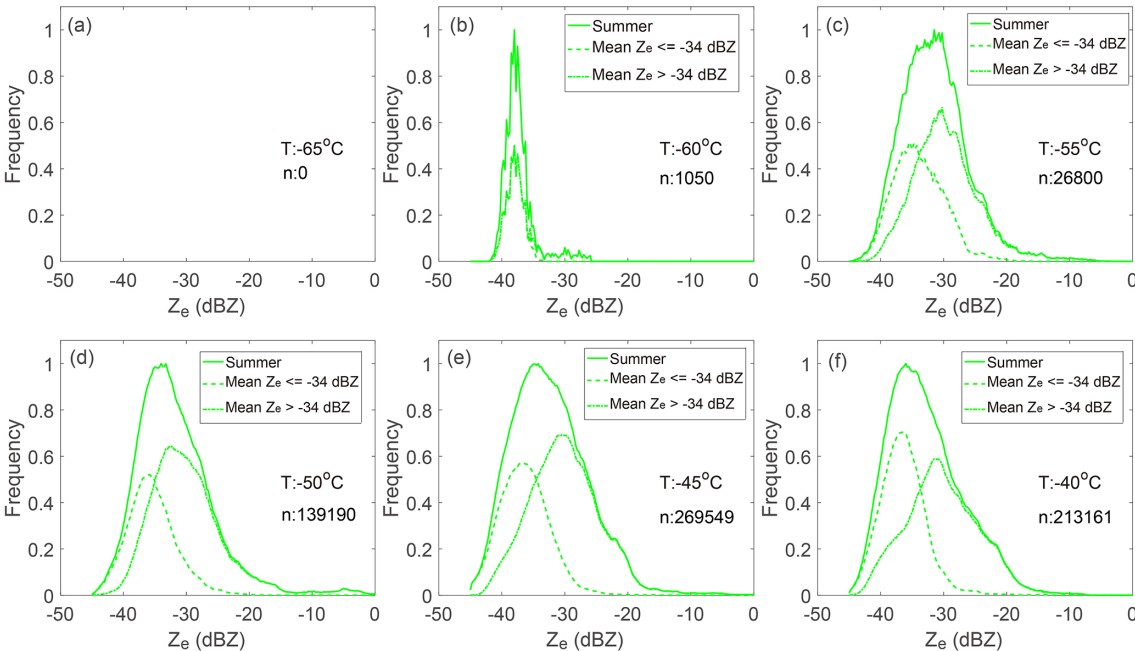

**Figure 9.** As in Fig. 8 but for summer. No cirrus clouds are found when temperature is at or below $-65\,°C$ in summer.

backscattering from numerous particles and the $Z_e$ is more sensitive to the larger particles in a cloud target.

We divided cirrus clouds into two groups using a mean-$Z_e$ threshold to study whether cirrus clouds over Beijing also originate from different mechanisms. If the PDFs of the two separate groups exhibit distinct features, it is possible that they form from different mechanisms. It is found that the cirrus clouds in spring and summer (Figs. 8 and 9) can be separated clearly into two groups by a threshold of $-34$ dBZ, and the two groups demonstrate different PDFs after apply-

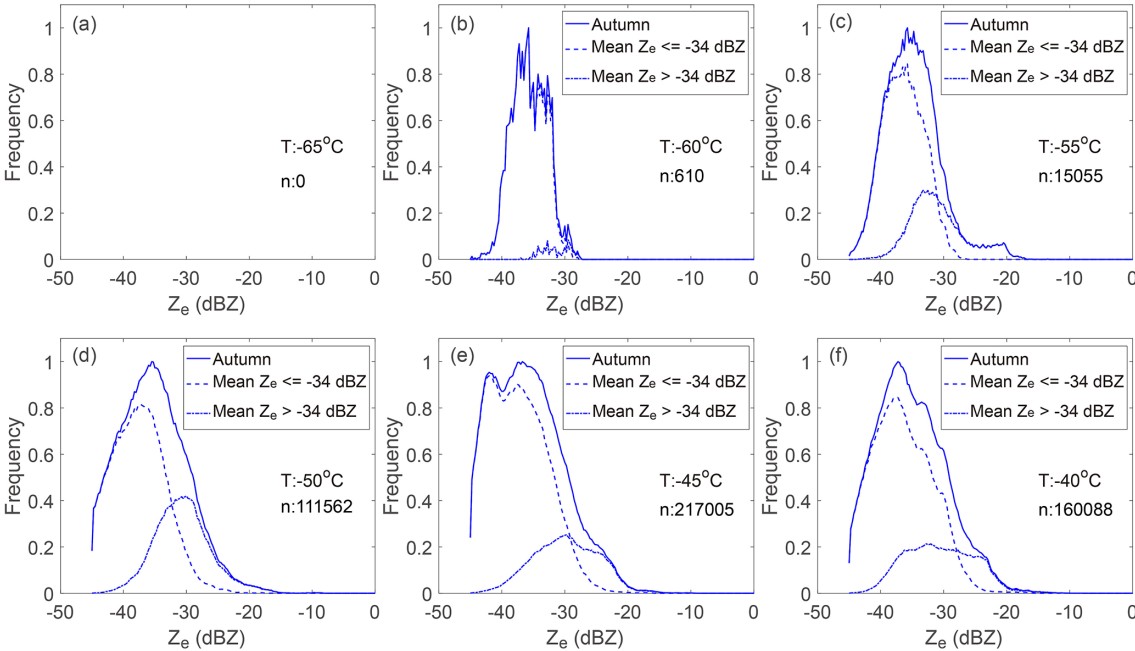

**Figure 10.** As in Fig. 8 but for autumn. No cirrus clouds are found when the temperature is at or below $-65\,^{\circ}$C in autumn.

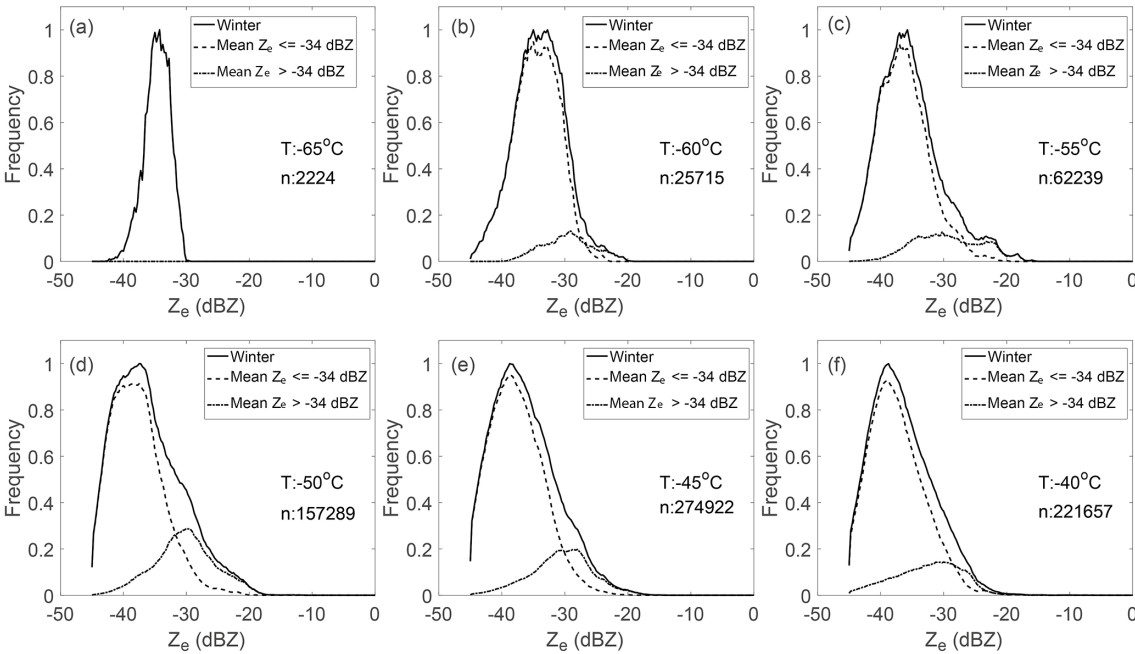

**Figure 11.** As in Fig. 8 but for winter.

ing each threshold between $-32\,$dBZ and $-38\,$dBZ. The full width at half maxima and the peak center are different. In addition, the proportions of the two groups are comparable. However, in autumn and winter, cirrus clusters with a mean reflectivity of less than $-34\,$dBZ contribute the absolute majority of all cirrus clusters when compared with the cirrus clusters with a reflectivity larger than $-34\,$dBZ, illus-

trating different PDFs from those in spring and summer. It is possible that the differences in the PDFs among the four seasons are due to the different origin types. For a mean $Z_e \leq -34\,$dBZ, a cirrus cloud is likely to be an in situ-origin cirrus cloud; for a mean $Z_e > -34\,$dBZ, a cirrus cloud is likely to be a liquid-origin cirrus cloud. From Figs. 8 and 9, it can be seen that spring has more in situ-origin cirrus clouds,

while summer has more liquid-origin cirrus clouds. In winter and autumn, cirrus clouds with a mean $Z_e \leq -34$ dBZ dominate, which means that the dominant cirrus clouds in winter and autumn are in situ-origin cirrus clouds. Summer has more convective movements and water vapor resulting in dominant liquid-origin cirrus clouds.

As mentioned above, there might be another possibility, which is that only one origin type dominates over Beijing, since most PDFs are unimodal. Large-scale synoptic and dynamic analysis should be carried out to distinguish the dominant origin type. At present, however, we prefer the view that there are two origin types in Beijing, since this is consistent with the basic weather characteristics in the four seasons. Nonetheless, more work will be performed in the future to confirm the current assumptions. On the whole, the formation mechanisms of cirrus clouds in spring and summer in Beijing illustrate different features to those in autumn and winter. It can also be found that the distribution of reflectivity depends not only on the temperature but also on the origin type.

## 6 Summary and discussion

Ice clouds are an important component of the planetary radiation budget and remain an uncertainty source in GCMs. This study used 4 years of vertically pointing Ka-band radar measurements in Beijing to characterize the physical and optical properties of ice clouds and to investigate the origination type of cirrus clouds. The goal was to present the quantified properties of ice clouds over the subtropical monsoon zone, which can be represented in GCMs to move toward a better understanding of the relationships between temperature and radar reflectivity under different formation conditions in various monsoon climates.

The winter monsoon and summer monsoon prevail alternately over Beijing, resulting in four distinct seasons. Ice clouds in winter and summer show strikingly different features. The specific findings about the properties of ice clouds can be summarized as follows:

1. The occurrence frequency, height, temperature and mean reflectivity of ice clouds in winter are lower than in summer. The average occurrence frequency over Beijing is 14 %, and it is 20 % in summer but less than 10 % in winter. The diurnal variation in the occurrence frequency is not obvious, indicating an insensitive response to solar heating.

2. The CTHs of ice clouds range within 5.5–12.9 km, and the difference between the maximum and minimum reaches 6 km in every season. The mean CTH in summer is 2.2 km higher than in winter. The CBHs range within 5–12.4 km, and the difference in the mean CBH between summer and winter is 2.1 km. In total, 86 % of ice clouds are above 7 km in summer and 81 % are above 7 km in winter. Statistically, in the four seasons, 68 % of clusters have a depth of less than 0.5 km, 90 % have a depth of less than 1 km, and 96 % have a depth of less than 1.5 km.

3. The EXT ranges through orders of magnitude from low values of less than 0.1 km to over 2800 km. Summer has the minimum mean, median and trimmed mean EXT, whereas ice clouds in autumn have the maximum mean, median and trimmed mean EXT. Statistically, about 75 % of ice clouds have an EXT of less than 50 km and 87 % have an EXT of less than 100 km. In addition, the mean COD in spring, summer, autumn and winter is 4.3, 6.1, 4.6 and 4.5, respectively.

4. The radar reflectivity of ice clouds is dependent on the height, temperature and CD. The reflectivity mostly varies between $-35$ and $-10$ dBZ, and the mean reflectivity in summer is 10 dBZ higher than in winter. More than 95 % of ice bins are below the temperature of $-15$ °C, and the mean temperature of ice cloud in winter is the lowest among the four seasons. It was found that there is a strong linear relationship between the mean reflectivity and the CD.

5. Cirrus cloud occurs more frequently at $-45$ °C than at other temperatures over Beijing, and cirrus clouds in spring are the most frequent among the four seasons.

The PDFs of reflectivity for cirrus cloud with respect to various temperatures were also investigated. It was found that the PDFs in the four seasons illustrate striking differences. A preliminary analysis indicates that cirrus clouds with mean reflectivity lower than $-34$ dBZ are likely to be of the in situ-origin type. Most cirrus clouds are of the in situ-origin type in winter and autumn; the in situ-origin cirrus clouds are more frequent than liquid-origin cirrus clouds in spring, while summer features more liquid-origin cirrus clouds. It should be noted that the current analysis and results might have limitations due to the KPDR's limited ability to identify possible supercooled layers in clouds. In our recent work (Huo et al., 2020), the cirrus clouds are separated into three types via cloud-base temperature to study their particle reflectivity and movements, since there are differences among previous studies in the knowledge of the temperature range of cirrus clouds. Besides seasonal variation, cirrus clouds with different cloud-base temperatures also have different microphysical characteristics. In future work, we intend to further investigate the formation mechanisms of cirrus clouds in Beijing, as well as in other areas, for the purposes of parameterization in GCMs and the development of a locally adaptive $Z_e$–IWC relationship.

*Data availability.* The ERA5 hourly data on pressure levels from 1979 to the present are available through the Copernicus Climate

Change Service (https://doi.org/10.24381/cds.bd0915c6, Hersbach et al., 2018). The AHI cloud property research product (produced from Himawari-8) that was used in this paper was supplied by the P-Tree System, Japan Aerospace Exploration Agency (https://www.eorc.jaxa.jp/ptree/userguide.html; Bessho et al., 2016.). The radar data used here are available by special request to the corresponding author (huojuan@mail.iap.ac.cn).

*Author contributions.* JH designed the study and carried it out. YT, CH, XW, YB, DL, SD and BL prepared some of the datasets. JH prepared the manuscript with contributions from all co-authors.

*Competing interests.* The authors declare that they have no conflict of interest.

*Acknowledgements.* We appreciate the valuable suggestions and insightful instructions from the reviewers. We also thank the ECMWF ERA5 and AHI science teams for sharing their product datasets. We also acknowledge our Ka-radar team for their maintenance service during long-term measurements that made our research possible.

*Financial support.* This research has been supported by the National Natural Science Foundation of China (grant no. 41775032).

*Review statement.* This paper was edited by Martina Krämer and reviewed by two anonymous referees.

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

**Remarks from the typesetter**

TS1    Please excuse me if my remark was unclear. It is possible to insert this change, however, we need the editor to approve it; after publication, these post-review adjustments will be visible in the peer-review tab. In case you still wish to change "September" to "December", please provide an explanation which can be forwarded to the editor. Upon approval, we will make the appropriate changes. Thank you for your understanding.