# Peer review of "Properties of ice cloud over Beijing from surface Ka-band radar observations during 2014–2017"

_Atmospheric Chemistry and Physics, 2020_

## Referee Comment (RC1) · Anonymous Referee #2 · 8 May 2020

Review of

**Properties of mid-latitude cirrus cloud**
**from surface Ka-band radar observations during 2014-2017**

by Juan Huo et al.

This paper presents an analysis of ice cloud properties (height, optical depth and horizontal extent) and formation mechanisms (in-situ or liquid origin) based on four years of surface millimetre wavelength radar measurements in Beijing, China. The results are proposed to serve as a reference for parameterization and characterization in global climate models.

I was also referee during the access review phase of the paper and recommended to consider some points before uploading the manuscript to ACPD, though they go in the direction of scientific comments. This review takes up this points again (it is an answer to the authors' response) since I am not content with the authors arguments and the changes they have made in the manuscript, as you will see in the new comments.

The original, first review of the manuscript is shown in **black**, the author's answers in blue and referees new responses in green.

Overall I find the paper a well-made study on ice clouds, the four-year data set allows a comprehensive insight into the appearance of the clouds at the measurement site; so far there are not many observations from this region. The paper is well written, the illustrations are clear, the topic is appropriate for ACP.

Nevertheless, I would recommend a change to the manuscript before it appears in ACPD, which I describe in the following.

**1)** The description of the Ka-band radar is very brief. I miss information of the upper and lower detection limits, i.e. the detection range, as well as uncertainties. This should be placed in the context of the corresponding range of the observed variables in cirrus clouds, so that it can be seen whether the complete range of the cirrus is covered by the measurements.

**1a)** Sorry for that. More details about the Ka radar can be found in the paper of Huo et al. 2019∗ . Since there are detailed descriptions, we introduce it briefly in this manuscript. According to referee's comments, we added the descriptions about the capability of the Ka radar (KPDR) (Please see the 1st paragraph in section 2.2).

I guess you meant the new text in Section 2.1 (not 2.2):  *'It should be noted that KPDR is insensitive to very small particles and it is possible that KPDR will miss some clouds with reflectivity out range of the detection threshold. The missed percentage is inaccessible at present due to our incomplete understanding of cirrus clouds and limitations of observation condition; however, it should be small according to the radar capability.'*

**New comments (A):**
First, I want to mention in general that it would have been good to show the new text in the answer so that the referee does not have to - time consumingly - compare the two paper versions to find out what has changed. I recommend this method for the next time.

Second, the answer is not very informative. The detection threshold (lower limit) should be given here, even if they are mentioned in earlier papers. It could not be expected from the reader to read other papers to find important information.

Further, if you state that 'The missed percentage (of clouds) is inaccessible' you cannot conclude that 'it should be small'. The conclusion would be that it is not known.

I insist on this point because to my opinion, for studies presenting general properties of clouds derived from a large data set from one instrument, then claiming that the observations can serve as a reference for parameterization and characterization in global climate models, it is essential to have a good knowledge if all occuring clouds are detected or not, and in case not, which part of the clouds are represented by the measurements.

– Couldn't it be possible to estimate from the definition of the radar reflectivity

$$Z = \int_0^\infty D^6 N_V(D)\, dD,$$

which cirrus clouds are most probably out of the detection range of the Ka radar with the knowledge of the size distributions? Cirrus cloud size distributions in different temperature intervals are recently shown by Sourdeval et al. (2018), ACP, their Figure 2. These size distributions could be used to estimate Z, and I would highly recommend to perform such calculations.

– There is a recent study (Jiang et al. (2019): Simulation of Remote Sensing of Clouds and Humidity From Space Using a Combined Platform of Radar and Multifrequency Microwave Radiometers, https://agupubs.onlinelibrary.wiley.com/doi/epdf/10.1029/2019EA000580; see also the EGU contribution: https://presentations.copernicus.org/EGU2020/EGU2020-20864_presentation.pdf ) detecting ice clouds with remote sensing methods from space, where the sensitivity range of the instuments is stated. Isn't such an analysis also possible for the ground based radar used here?

1b) Also, we added some statements (→i.e. cirrus cloud definitions) to make the information clearer (Please see the second paragraph in section 2.2).

Your new text: *'In some studies (Krämer et al. 2016; Luebke et al. 2016; Heymsfield et al. 2017; Wolf et al. 2018), cirrus clouds are defined as ice clouds with lower temperature < -38°C. In this study, according to the Glossary of the American Meteorological Society (AMS, 2019), the cirrus clouds are referred to all types of cirriform clouds (Ci, Cc and Cs clouds), which is determined by the reflectivity, temperature, height and depth.'*

**New comment (B):** I do not agree with the idea that cirrus clouds can be defined so differently. The definition of Ci, Cc and Cs clouds also includes an altitide range, namely roughly > 5 km. Also, cirrus clouds are characterized as detached, thin ice clouds.

Ac and As are reported up to altitudes of roughly 7 km; they are much thicker spatially  and also optically and are mostly completely glaciated at higher altitudes.
At mid-latitudes, the altitude range 5-7km corresponds to temperatures between about  -25 to -35 C (see Luebke et al., 2013, ACP, Fig. 4), where most of the ice clouds are glaciated mixed phase clouds, some might be fall streaks of cirrus clouds from above.

**B-1**: I strongly suggest that the authors reconsider the definition of the clouds they have observed. I recommend to define (throughout the manuscript, and already in the title) that ice clouds in the temperature range -15 to -55 C  are detected  (i.e. ice clouds in the mixed phase as well as the warmer part of cirrus cloud range).  See also my new comments on 2).

**B-2**: I further suggest to change the title of the paper to

‚Properties of mid-latitude  **ice clouds** from surface Ka-band radar
  observations during 2014-2017'

1c) The KPDR can detect the target of which reflectivity larger than -45 dBz due to its stronger transmitter, illustrating higher capability relative to other Ka radar, i.e., Ka radars with all-solid transmitter generally use reflectivity threshold of no more than -40 dBz. For the volume reflectivity, large particles normally contribute more than small particles. KPDR will miss small particles of which reflectivity lower than the -45 dBz. At present, it is hard to estimate the missed portions due to shortage of comparable measurements.

See my new comment (**A)**

2) This is of importance, because in the study cirrus clouds are reported only down to temperatures of -45 C, which is too high for cirrus clouds. Cirrus cover the temperature range down to about – 70 C at mid-latitudes and to about -90 C in the tropics (see e.g. Schiller et al., 2008, JGR; Luebke et al., 2013, ACP).

Further,  cloud observations up to temperatures of -20 C are reported, which is definitively out of the cirrus temperature range. In newer studies, as for example Krämer et al. (2016, ACP), Luebke et al. (2016, ACP), Heymsfield et al. (2017, AMS Met. Monographs), Wolf et al. (2018, ACP; 2019, GRL), Krämer et al. (2020, ACPD), cirrus clouds are defined as ice clouds in the temperature range < -38 C, while in the range between -38 and 0 C mixed phase clouds (liquid and/or ice) occur.  The sorting of cirrus in the types of in situ and liquid origin (that have been succesfully used in the study – these are very good results!) is based on this temperature criterion, and, consequently it is found in the paper that at > -38 C the clouds are liquid origin.  This shows on the one hand that the reflectivity based cloud origin sorting works well, but on the other hand that the chosen definition of cirrus is not suitable here. From my point of view, in this study clouds in the mixed phase range* as well as the warmer part of cirrus** are detected.

Thank the referee very much for the considered comments.
Yes. We have noticed that definitions of cirrus clouds differ among various references. In this study, cirrus clouds are defined and determined from the reflectivity distribution, height, depth and temperature. The classification accuracy is 86% when compared with meteorological observer (Huo et al. 2019). After two temperature criteria (the cloud-top temperature should

be less than −30°C and the cloud-base temperature should be less than 0°C added), about 15% cirrus clusters are filtered (from 7750 to 6649). Cirrus clouds in this study occur mostly within temperatures range of −15oC to −55oC, and are referred to all types of cirriform clouds (Ci, Cc and Cs clouds). If -38°C is used as criteria for cirrus determination, then about 60% cirrus clusters will not be considered. The occurrence frequency will reduce to 5% which is obviously different to the real distribution.
→ see new comment (**B**)

Does the cirrus lower than -70°C exist over Beijing? Or they are missed by the KPDR?
It is very likely that cirrus clouds down to -70C exist above Beijing at mid – latitudes, because small and large scale temperature fluctuations will be present as in other regions. In this study, the coldest  detected  cirrus temperatures  are -55C. The fact that cirrus clouds are getting thinner (with smaller ice crystals) at colder  temperatures together with the knowledge that smaller ice crystals can not be detected by Ka band radar makes it very likely that the thinnest, highest and  coldets cirrus are missed. → see also new comment (**A**)
Or cirrus cloud should be warmer than -38°C? → see new comment (**B**)  More investigations are required to answer → I think we already have answers! . It was found that the temperature of cirrus clouds over North China from the CALIPSO/CloudSat 2C-ICE products were also far warmer than -38°C and above 98% of those cirrus clouds are ice particles (Fig.1, Huo et al. 2014※) → I would call them ice clouds, see new comment (**B**). In addition, the origination type analysis in section 5 also show consistent features to the cirrus lower than -38°C.
**New comment (C):** As I pointed out in 2)  it is found in the paper (Section 5) that at > -38 C the clouds are liquid origin. Liquid origin cirrus are defined as glaciated mixed phase clouds that are lifted to altitudes where the temperature is < –38C. Consequently, if you find ice clouds with this characteristics at temperatures  > -38C, these are glaciated mixed phase clouds.
Then, the definition in this study might be reasonable and we hope referee can accept our current cirrus classification approach. → see new comment to (**B**)

*:  Can thick clouds also be detected with the Ka band radar or is there an upper limit ?

KPDR can detect thick clouds and clouds with depth larger than 10 km had been measured in past days (see a case in Fig.2).

**: the colder, thin cirrus are below the radar detection limit, yes ?

Theoretically, it is possible that the Ka radar will miss cloud bins with low number condensation and very small particles resulting in very small volume reflectivity. If colder and thin cirrus contain a few small particles, it may be missed by KPDR. At present, it is hard to estimate the missed percentage because there are no other comparable data sets at Beijing. It is hoped to get some achievements in the future as we have a lidar and Ka radar making coincident measurements at Tibet and Lidar is more sensitive to small particles.
→ see new comment to (**A**)

[Figure]

Fig.1 cited from Huo et al. 2014

※Huo, Juan and Daren Lu. 2014: Physical properties of high-level cloud over land and ocean from CloudSat/CALIPSO data. Journal of Climate, Vol. 27, No. 23. 8966-8978.

[Figure]

Fig. 2

3) Though this is already a part of the scientific discussion that usually takes place only in the open discussion, I recommend to consider these points already before uploading the paper to ACPD.

Yes. We agree. →however, I do not see that the suggested changes were made.

---

## Referee Comment (RC2) · Anonymous Referee #1 · 10 May 2020

Review of 'Properties of mid-latitude cirrus cloud from surface Ka-band radar observations during 2014-2017' by Huo et al.

This manuscript by Huo et al. shows a climatological analysis of ice cloud macrophysical properties based on measurements of a ground-based radar situated in Beijing, China. More specifically, the authors explore four years of radar measurements, from which are identified cirrus clusters, and statistically analyse the occurrence, cloud heights, depths and horizontal extents associated with these clusters. Microphysical retrievals are not performed but coincident cloud optical depths by Himawari-8 are used to complete the analyses. Finally, the relation between reflectivity distribution and cirrus origin is investigated.

The manuscript is rather clear and well written. This study is motivated by the large uncertainties on our current understanding of ice cloud formation mechanisms and their representation in climate models, which I agree constitute significant problems. However, I strongly feel that two points hold this manuscript back in that effort and it was quite difficult to view past them for the review. First, the definition of cirrus used by the author must be better clarified. Second, the instrumental sensitivity to the said cirrus should be quantified. Further details on these two points are provided below. Unfortunately, without providing this precise context to the study, I'm afraid the manuscript and its findings cannot be as useful to the community as they should be. There is otherwise a lot of potential in this radar dataset and the analyses performed here, particularly concerning the observed seasonal variations of cloud properties and in the cirrus origin study that is new and very interesting.

I therefore strongly encourage the authors to clarify the manuscript by responding to the following comments, and advise the editor to accept this manuscript only after substantial major revisions.
* * *
General comments:

1. My two general comments are actually related. The first one concerns the cirrus classification. The identification procedure described in section 2.2 is twofold. First, clusters are categorised in nine cloud types following a procedure described in a procedure detailed in a previous study by the first author, and two types are selected. It would still be very useful to have a short summary of the main features of the "Cs" and "Cc" classes in this section. Then the cloud-top (CTT) and cloud-base (CBT) temperatures are used to filter out cirrus clouds. The conditions are CTT < -30°C and CBT > 0°C. I understand that this can be highly debated but considering the instrumental setup (see point 2) in this study I think it would be wiser to go for the stricter CTT < -40°C threshold and avoid strong contamination by mixed-phase layers. I might have a biased perspective but when reading the title I'd expect a study about "cold" ice clouds (and their associated processes), while many of the results focus on in-cloud temperatures > -38°C so it gives the (perhaps wrong) impression that these are largely represented in the dataset. If it is decided to keep the same threshold then an histogram of the CTT and in-cloud temperature distributions must at least be presented so that the types of cirrus explored in this study are made clear to the readers.

On a side note, I do not understand why the authors use the Heymsfield et al. (2017) reference to justify the -30°C threshold while including again this reference when explaining that other study use a -38°C threshold, please be consistent.

2. More importantly, and I think critical for this paper, is the question of the instrumental sensitivity. In section 2.2 the authors remind us that cirrus are typically associated with very small optical depths (e.g. OD < 3, in this section), and one can logically wonder how many of these clouds are actually detected by the ground-based radar. And even if detected, what portions of these cirrus are actually seen in terms of vertical and horizontal extents? This has critical impacts on the determination of the cloud-top height, occurrence and horizontal extents, which are central results to the study, and must

be answered to properly set the context of the study. At minimum, it is necessary to show a sensitivity analysis of this cloud radar to the cirrus optical depth (or ice water content).

Such preliminary analysis is presented in section 3.4 where the Himawari-8 COD retrievals are compared to the radar reflectivity measurements, but these have a few issues. First, CODs up to 25 are detected, how do these classify as cirrus? Second, the Himawari-8 product used here is based on visible / near-IR channels and it is well known that these are not the retrieval of choices for thin cirrus studies, where thermal infrared channels should be preferred. There seems to be Himawari-8 retrievals of ice cloud properties that use such split-window techniques (e.g. Iwabuchi et al, JMSJ, 2018) – these would be much more robust and meaningful. Additionally to the analyses currently shown in section 3.4 the authors should show distributions of Himawari-8 COD and CTH retrievals when the radar detects a cirrus but also when it doesn't, to estimate the amount of missed cirrus and biases in their CTH due to the radar sensitivity. Another possibility would be to use lidar measurements from the same region to detect the cloud occurrence, if any is available to the authors.

3. On a more positive note, the two first comments do not impact the validity of some of the findings such as the seasonal variations (since the same detection threshold applies), or the origin analysis that are based on the reflectivity. There are very interesting finding worth being published and I'd be very much willing to discuss them further after proper clarification of the two aforementioned points in the revised manuscript.
* * *
Specific comments:

Considering the importance and impact that revising the first two points might have on the paper, I have for now limited specific comments to a couple of important points directly related to them.

1. The introduction is very general, it would be more interesting to discuss a bit more the state-of-the-art in radar-only cirrus analyses. How did they tackle the instrumental sensitivity issues to provide usable climatologies?

2. Regarding the occurrence analyses (section 3.2), could you please comment on cirrus detection in the presence of underlying liquid clouds or precipitation? Would that be a limiting factor to the statistics presented here?

3. In section 3.4 and after, a "mean reflectivity" is used. Could you please define the meaning of this quantity? Is it vertically averaged?

---

## Author Comment (AC1) · 27 May 2020

**Properties of mid-latitude cirrus cloud from surface Ka-band radar observations during 2014-2017**

by Juan Huo et al.

This paper presents an analysis of ice cloud properties (height, optical depth and horizontal extent) and formation mechanisms (in-situ or liquid origin) based on four years of surface millimetre wavelength radar measurements in Beijing, China. The results are proposed to serve as a reference for parameterization and characterization in global climate models.

I was also referee during the access review phase of the paper and recommended to consider some points before uploading the manuscript to ACPD, though they go in the direction of scientific comments. This review takes up this points again (it is an answer to the authors' response) since I am not content with the authors arguments and the changes they have made in the manuscript, as you will see in the new comments.

The original, first review of the manuscript is shown in **black**, the author's answers in blue and referees new responses in green.

We are very sorry that our last answers did not satisfy reviewer. Thanks to reviewer for giving us the opportunity to answer them again. The problems are focused on two aspects, one is the detection capability of Ka radar (the comment A), and the other is the selection criteria or definition of cirrus (the comment B). Our replies to the new comments are presented in orange. Revised portions, including all figures, in the revised manuscript are marked with red. We hope reviewer could be content with our new answers and revisions.

Overall I find the paper a well-made study on ice clouds, the four-year data set allows a comprehensive insight into the appearance of the clouds at the measurement site; so far there are not many observations from this region. The paper is well written, the illustrations are clear, the topic is appropriate for ACP.

Nevertheless, I would recommend a change to the manuscript before it appears in ACPD, which I describe in the following.

**1)** The description of the Ka-band radar is very brief. I miss information of the upper and lower detection limits, i.e. the detection range, as well as uncertainties. This should be placed in the context of the corresponding range of the observed variables in cirrus clouds, so that it can be seen whether the complete range of the cirrus is covered by the measurements.

**1a)** Sorry for that. More details about the Ka radar can be found in the paper of Huo et al. 2019 . Since there are detailed descriptions, we introduce it briefly ※ . Since there are detailed descriptions, we introduce it briefly in this manuscript. in this manuscript. According to referee's comments, we added the descriptions about the capability of the Ka radar (KPDR) (Please see the 1st paragraph in section 2.2).

I guess you meant the new text in Section 2.1 (not 2.2): *'It should be noted that KPDR is insensitive to very small particles and it is possible that KPDR will miss some clouds with reflectivity out range of the detection threshold. The missed percentage is inaccessible at present due to our incomplete understanding of cirrus clouds and limitations of observation condition; however, it should be small according to the*

*radar capability.'*

**New comments (A):**

First, I want to mention in general that it would have been good to show the new text in the answer so that the referee does not have to - time consumingly - compare the two paper versions to find out what has changed. I recommend this method for the next time.

Second, the answer is not very informative. The detection threshold (lower limit) should be given here, even if they are mentioned in earlier papers. It could not be expected from the reader to read other papers to find important information.

Further, if you state that 'The missed percentage (of clouds) is inaccessible' you cannot conclude that 'it should be small'. The conclusion would be that it is not known. I insist on this point because to my opinion, for studies presenting general properties of clouds derived from a large data set from one instrument, then claiming that the observations can serve as a reference for parameterization and characterization in global climate models, it is essential to have a good knowledge if all occuring clouds are detected or not, and in case not, which part of the clouds are represented by the measurements.

– Couldn't it be possible to estimate from the definition of the radar reflectivity

$$Z = \int_{D_{min}}^{D_{max}} D^6 N_v(D) dD \quad (1)$$

which cirrus clouds are most probably out of the detection range of the Ka radar with the knowledge of the size distributions? Cirrus cloud size distributions in different temperature intervals are recently shown by Sourdeval et al. (2018), ACP, their Figure 2. These size distributions could be used to estimate Z, and I would highly recommend to perform such calculations.

– There is a recent study (Jiang et al. (2019): Simulation of Remote Sensing of Clouds and Humidity From Space Using a Combined Platform of Radar and Multifrequency Microwave Radiometers, https://agupubs.onlinelibrary.wiley.com/doi/epdf/10.1029/2019EA000580; see also the EGU contribution: https://presentations.copernicus.org/EGU2020/EGU2020-20864_presentation.pdf) detecting ice clouds with remote sensing methods from space, where the sensitivity range of the instuments is stated. Isn't such an analysis also possible for the ground based radar used here?

We understand that the detection capability of KPDR determines the quality of this study. We are very grateful to reviewer for providing us two options for testing the detection capabilities of KPDR. The first scheme calculates Ze based on the particle scattering theory combined with the knowledge of particle size distribution. The second scheme calculates Ze using a cloud model. The second scheme requires input cloud field data which should contain information of cloud micro-physical parameters such as cloud particle size and IWC, as so on. Relatively, the parameter conditions required by the first scheme are easier to meet, and its calculation accuracy is also sufficient to estimate the radar detection capability. Therefore, we decide to use the first option.

-- We first consider the extreme condition, assuming that the cloud particle size in

a scattering target (radar bin) is a constant, then, calculate the Ze in terms of various Ni according to the equation (1) provided above by reviewer (see Fig. RC1).

[Figure]

Fig. RC1. The reflectivity factor (Ze) in terms of and number density (Ni) when particle diameter (D) is a constant.

The lowest detectable value of KPDR is - 45dBZ. Generally, it is considered that the Ze of a scattering bin less than this value cannot be observed, otherwise it can be detected if Ze is above -45 dBZ. Taking the blue dotted horizontal line (Ze = -45dBZ) on the Fig.RC1 as a reference, under extreme conditions, when the particle size is equal to 20um, the Ze is less than -45dBZ even if the Ni is greater than 500 / L, meaning that KPDR cannot detect them. However, for the particles of which size is equal to 60um, KPDR can detect them only if Ni is larger than 1 / L. It can be concluded that that radar is more sensitive to large particles than small particles.

--Actually, the particle size of natural ice cloud is not constant, and the distribution function is much more complex than the above assumption, which is generally expressed by a modified gamma function. In addition, the Ze from ice particles is also related to the shape of particles. Deng et al. (2010, Fig. 4) simulated and calculated the distribution of Ze for several different shapes of ice crystals based on a modified gamma function and a particle size distribution (PDF) obtained by *in-situ* detections over mid-latitude region, respectively. For the convenience of reviewers, the relevant parts in Deng et al. (2010, Fig. 4) are posted here and shown as Fig.RC2:

[Figure]

Fig.RC2 Left is the reflectivity (Ze) of five types of ice crystals calculated using modified the

gamma function as the particle size function. Right is the reflectivity of four types of ice crystals calculated using the PDF obtained by *in-situ* measurement.

It can be seen from Fig.RC2 that the simulated reflectivity is above -30 dBZ. In addition, Deng et al. (2010, Fig. 11) also presented a comparison of detection ability between the CloudSat/CPR (radar) and CALIPSO/CALIOP (lidar). In order to facilitate reviewers, we also extracted the figure which is shown as Fig.RC3 below. Because the detection sensitivity of CPR is about -30 dBZ, a small part of the thin cloud observed by CALIOP is missed by CPR. The reflectivity of these "missed" thin clouds is approximately between -30 and -45dBZ.

[Figure]

Fig. RC3  Retrieval example of 22 July 2007 CloudSat-CALIPSO case during the TC4 experiment. (a, b) Measured and simulated radar reflectivity, respectively. (c, d) Measured and simulated lidar back-scattering, respectively.

--Pokharel et al. (2011) compared the measured 94-GHz radar reflectivity in midlevel ice clouds with the reflectivity calculated using particle size distribution determined with a particle imaging probe. Below Fig.RC4 is cited from the figure 7 of Pokharel et al. (2011).

[Figure]

Fig.RC4 Scatterplots of the measured reflectivity $Z_{obs}$ vs the calculated reflectivity for 1-Hz data. (left) The d1 density assumption. (right) The d3 density assumption (dark points).

--In the work of Martrosov et al. (2017, Fig.1),for those ice hydrometeor collected during aircraft flights in midlatitude (left) and subtropical (right) regions,the reflectivity is generally above -25 dBZ (shown below as Fig.RC5).

[Figure]

Fig.RC5 Scatterplots of $K_a$-band radar reflectivity vs (a),(b) the PSD slope parameter $\Lambda$ and (c),(d) median volume particle size as derived from the (left) GCPEX and (right) CRYSTAL-FACE microphysical data.

Therefore, according to the calculations and measurements in previous and recent researches, KPDR should have strong capability in detecting most ice clouds in nature since its sensitivity is -45 dBZ.

Deng, M., G. G. Mace, Z. Wang, and H. Okamoto: Tropical composition, cloud and climate coupling experiment validation for cirrus cloud profiling retrieval using CloudSat radar and CALIPSO lidar, J. Geophys. Res. Atmos., 115, D00J15, doi:10.1029/2009JD013104,2010.

Pokharel, B. and G. Vali, Evaluation of collocated measurements of radar reflectivity and particle sizes in ice clouds: Journal of Applied Meteorology and Climatology, 2011. 50: p. 2104-2119.

Matrosov, S.Y. and A.J. Heymsfield, Empirical relations between size parameters of ice hydrometeor populations and radar reflectivity. Journal of Applied Meteorology and Climatology, 2017. 56(9): p. 2479-2488.

1b) Also, we added some statements (→i.e. cirrus cloud definitions) to make the information clearer (Please see the second paragraph in section 2.2).
Your new text: '*In some studies (Kr et al. 2016; Luebke et al. 2016; Heymsfield et al. 2017; Wolf et al. 2018), cirrus clouds are defined as ice clouds with lower temperature < -38°C. In this study, according to the Glossary of the American Meteorological Society (AMS,2019), the cirrus clouds are referred to all types of*

*cirriform clouds (Ci, Cc and Cs clouds),which is determined by the reflectivity, temperature, height and depth.'*

**New comment (B):**

I do not agree with the idea that cirrus clouds can be defined so differently. The definition of Ci, Cc and Cs clouds also includes an altitude range, namely roughly > 5 km. Also, cirrus clouds are characterized as detached, thin ice clouds. Ac and As are reported up to altitudes of roughly 7 km; they are much thicker spatially and also optically and are mostly completely glaciated at higher altitudes.

At mid-latitudes, the altitude range 5-7km corresponds to temperatures between about -25° to -35°C (see Luebke et al., 2013, ACP, Fig. 4), where most of the ice clouds are glaciated mixed phase clouds, some might be fall streaks of cirrus clouds from above.

**B-1**: I strongly suggest that the authors reconsider the definition of the clouds they have observed. I recommend to define (throughout the manuscript, and already in the title) that ice clouds in the temperature range -15 to -55°C are detected (i.e. ice clouds in the mixed phase as well as the warmer part of cirrus cloud range). See also my new comments on 2).

**B-2**: I further suggest to change the title of the paper to,Properties of mid-latitude **ice clouds** from surface Ka-band radar observations during 2014-2017'

There are different versions of cirrus clouds definition in previous articles. We initially thought that their differences may be due to the different scope of the definition, some may be a broad definition and the other may be a narrow definition.

Reviewer mentioned that cirrus covered the temperature range down to about – $70^{o}$C at mid-latitudes. We agree that. However, within mid-latitude regions, regional difference in temperature distribution should be considered. The coverage range of temperature would have local characteristics.

In the figure below (Fig.RC6), all the temperature profiles obtained by sounding measurements and downloaded from the hourly ERA5 dataset in Beijing in 2015 are presented. The dotted lines indicate the -70 $^{o}$C. It can be seen that most temperature within 15km is above -70 $^{o}$C, and -70$^{o}$C temperature occurs scarcely. Also, -65$^{o}$C temperature occupy very small percent. In the 5-7km altitude range, the temperature range is approximately -5 ~ -30 $^{o}$C in Beijing. Thus, Beijing presents a distinct temperature distribution. Ice clouds over Beijing demonstrate a distinct temperature range.

In order to avoid confusions and consider the suggestions of reviewers, we decide to adopt the opinions of reviewers and define those clouds analyzed in our study as ice clouds. And in the revised document, in order to ensure 100% ice cloud, we increased the temperature threshold standard, and the cloud-base temperature is required to be below -10 $^{o}$C. In addition, in the revised manuscript, the cirrus clouds are picked out from ice clouds according to cloud-base temperature less than -38 $^{o}$C for a contrast. The occurrence frequency (see section 3.2) and formation type of cirrus (section 5) are analyzed.

[Figure]

Fig.RC6 Temperature profiles at Beijing form radiosonde sounding (left) and ERA5 (right).

1c) The KPDR can detect the target of which reflectivity larger than -45 dBz due to its stronger transmitter, illustrating higher capability relative to other Ka radar, i.e., Ka radars with all-solid transmitter generally use reflectivity threshold of no more than -40 dBz. For the volume reflectivity, large particles normally contribute more than small particles. KPDR will miss small particles of which reflectivity lower than the -45 dBz. At present, it is hard to estimate the missed portions due to shortage of comparable measurements.
See my new comment (**A**)

2) This is of importance, because in the study cirrus clouds are reported only down to temperatures of -45˚C, which is too high for cirrus clouds. Cirrus cover the temperature range down to about – 70˚C at mid-latitudes and to about -90˚C in the tropics (see e.g. Schiller et al.,2008, JGR; Luebke et al., 2013, ACP).

Further, cloud observations up to temperatures of -20˚C are reported, which is definitively out of the cirrus temperature range. In newer studies, as for example Kramer et al. (2016, ACP), Luebke et al. (2016, ACP), Heymsfield et al. (2017, AMS Met. Monographs), Wolf et al. (2018, ACP; 2019, GRL), Kramer et al. (2020, ACPD), cirrus clouds are defined as ice clouds in the temperature range < -38˚C, while in the range between -38˚C and 0˚C mixed phase clouds (liquid and/or ice) occur. The sorting of cirrus in the types of in situ and liquid origin (that have been succesfully used in the study – these are very good results!) is based on this temperature criterion, and, consequently it is found in the paper that at > -38˚C the clouds are liquid origin. This shows on the one hand that the reflectivity based cloud origin sorting works well, but on the other hand that the chosen definition of cirrus is not suitable here. From my point of view, in this study clouds in the mixed phase range* as well as the warmer part of cirrus** are detected.
Thank the referee very much for the considered comments.
Yes. We have noticed that definitions of cirrus clouds differ among various references. In this study, cirrus clouds are defined and determined from the reflectivity distribution, height, depth and temperature. The classification accuracy is 86% when

compared with meteorological observer (Huo et al. 2019). After two temperature criteria (the cloud-top temperature should be less than −30°C and the cloud-base temperature should be less than 0°C added), about 15% cirrus clusters are filtered (from 7750 to 6649). Cirrus clouds in this study occur mostly within temperatures range of −15ºC to −55ºC, and are referred to all types of cirriform clouds (Ci, Cc and Cs clouds). If -38°C is used as criteria for cirrus determination, then about 60% cirrus clusters will not be considered. The occurrence frequency will reduce to 5% which is obviously different to the real distribution.

→ see new comment (**B**)

Does the cirrus lower than -70°C exist over Beijing? Or they are missed by the KPDR?

It is very likely that cirrus clouds down to -70°C exist above Beijing at mid – latitudes, because small and large scale temperature fluctuations will be present as in other regions. In this study, the coldest detected cirrus temperatures are -55C. The fact that cirrus clouds are getting thinner (with smaller ice crystals) at colder temperatures together with the knowledge that smaller ice crystals can not be detected by Ka band radar makes it very likely that the thinnest, highest and coldets cirrus are missed. → see also new comment (**A**)

Or cirrus cloud should be warmer than -38°C? →see new comment (**B**) More investigations are required to answer →I think we already have answers! . It was found that the temperature of cirrus clouds over North China from the CALIPSO/CloudSat 2C-ICE products were also far warmer than -38°C and above 98% of those cirrus clouds are ice particles (Fig.1, Huo et al. 2014[※]) . Since there are detailed descriptions, we introduce it briefly in this manuscript.) →I would call them ice clouds, see new comment (**B**). In addition, the origination type analysis in section 5 also show consistent features to the cirrus lower than -38°C.

[Figure]

FIG. 5 Frequency of IER with (top) heights and (bottom) temperatures over the (left) L and (right) O area during 2007–10.

**New comment (C):** As I pointed out in 2) it is found in the paper (Section 5) that at > -38°C the clouds are liquid origin. Liquid origin cirrus are defined as glaciated mixed phase clouds that are lifted to altitudes where the temperature is < –38°C. Consequently, if you find ice clouds with this characteristics at temperatures > -38°C, these are glaciated mixed phase clouds.

In the revised manuscript, cirrus clouds ($T_{base}$ < -38 ºC) are picked out from ice clouds and their origin types are investigated in the revised section 5. We have got

some new understandings, and we sincerely ask the reviewers to give some opinions about this part work.

Then, the definition in this study might be reasonable and we hope referee can accept our current cirrus classification approach. →see new comment to (**B**)

*: Can thick clouds also be detected with the Ka band radar or is there an upper limit ?

KPDR can detect thick clouds and clouds with depth larger than 10 km had been measured in past days (see a case in Fig.2).

[Figure]

**: the colder, thin cirrus are below the radar detection limit, yes ?

Theoretically, it is possible that the Ka radar will miss cloud bins with low number condensation and very small particles resulting in very small volume reflectivity. If colder and thin cirrus contain a few small particles, it may be missed by KPDR. At present, it is hard to estimate the missed percentage because there are no other comparable data sets at Beijing. It is hoped to get some achievements in the future as we have a lidar and Ka radar making coincident measurements at Tibet and Lidar is more sensitive to small particles.

→see new comment to (**A**)

3) Though this is already a part of the scientific discussion that usually takes place only in the open discussion, I recommend to consider these points already before uploading the paper to ACPD.

Yes. We agree. →however, I do not see that the suggested changes were made.

---

## Author Comment (AC2) · 27 May 2020

Review of 'Properties of mid-latitude cirrus cloud from surface Ka-band radar observations during 2014-2017' by Huo et al.

This manuscript by Huo et al. shows a climatological analysis of ice cloud macrophysical properties based on measurements of a ground-based radar situated in Beijing, China. More specifically, the authors explore four years of radar measurements, from which are identified cirrus clusters, and statistically analyse the occurrence, cloud heights, depths and horizontal extents associated with these clusters. Microphysical retrievals are not performed but coincident cloud optical depths by Himawari-8 are used to complete the analyses. Finally, the relation between reflectivity distribution and cirrus origin is investigated.

The manuscript is rather clear and well written. This study is motivated by the large uncertainties on our current understanding of ice cloud formation mechanisms and their representation in climate models, which I agree constitute significant problems. However, I strongly feel that two points hold this manuscript back in that effort and it was quite difficult to view past them for the review. First, the definition of cirrus used by the author must be better clarified. Second, the instrumental sensitivity to the said cirrus should be quantified. Further details on these two points are provided below. Unfortunately, without providing this precise context to the study, I'm afraid the manuscript and its findings cannot be as useful to the community as they should be. There is otherwise a lot of potential in this radar dataset and the analyses performed here, particularly concerning the observed seasonal variations of cloud properties and in the cirrus origin study that is new and very interesting.

I therefore strongly encourage the authors to clarify the manuscript by responding to the following comments, and advise the editor to accept this manuscript only after substantial major revisions.

Thanks to reviewer for the comments and encouragements. We have made careful revisions according to both reviewers' suggestions. Our responses here and below are marked in orange which follows the format of our replies to referee#2. Revision portions in the revised manuscript are marked with red. We hope reviewer can accept them.
* * *
General comments:

1. My two general comments are actually related. The first one concerns the cirrus classification. The identification procedure described in section 2.2 is twofold. First, clusters are categorised in nine cloud types following a procedure described in a procedure detailed in a previous study by the first author, and two types are selected. It would still be very useful to have a short summary of the main features of the "Cs" and "Cc" classes in this section. Then the cloud-top (CTT) and cloud-base (CBT) temperatures are used to filter out cirrus clouds. The conditions are CTT < -30ºC and CBT > 0ºC. I understand that this can be highly debated but considering the instrumental setup (see point 2) in this study I think it would be wiser to go for the stricter CTT < -40ºC threshold and avoid strong contamination by mixed-phase layers. I might have a biased perspective but when reading the title I'd expect a study about "cold" ice clouds (and their associated processes), while many of the results focus on in-cloud temperatures > -38ºC so it gives the (perhaps wrong) impression that these are largely represented in the dataset. If it is decided to keep the same threshold then an histogram of the CTT and in-cloud temperature distributions must at least be presented so that the types of cirrus explored in this study are made clear to the readers.

On a side note, I do not understand why the authors use the Heymsfield et al. (2017) reference to justify the -30ºC threshold while including again this reference when explaining that other study use a -38ºC threshold, please be consistent.

As mentioned by reviewer, there are different definitions for cirrus cloud in previous publications, some of which use the - 30 ºC, some of which use the - 10 ºC, and some use 0 ºC as a temperature threshold. We had thought that the difference may be due to the different scope of the definition, some are broad definitions and the other are narrow definitions. Our original definition standard mainly refers to the judgment standard from CloudSat/CPR, because the detection principle of CPR is similar to KPDR.

Considering the advices from two reviewers, we decided to adopt the opinions of reviewers and define these clouds as ice clouds in order to avoid confusion. And in the revised document, in order to ensure 100% ice cloud,

we raised the temperature threshold standard: cloud-base temperature is required to be below -10 $^o$C. Cirrus cloud is ice cloud. In the revised manuscript, the cirrus clouds are picked out from ice clouds according to cloud-base temperature ($T_{base} < -38$ $^o$C) for specific analysis and for a contrast. For example, the occurrence frequency (see section 3.2) and formation type of cirrus (section 5) are analyzed. We hope that our revisions will be approved by reviewer.

2. More importantly, and I think critical for this paper, is the question of the instrumental sensitivity. In section 2.2 the authors remind us that cirrus are typically associated with very small optical depths (e.g. OD < 3, in this section), and one can logically wonder how many of these clouds are actually detected by the ground-based radar. And even if detected, what portions of these cirrus are actually seen in terms of vertical and horizontal extents? This has critical impacts on the determination of the cloud-top height, occurrence and horizontal extents, which are central results to the study, and must be answered to properly set the context of the study. At minimum, it is necessary to show a sensitivity analysis of this cloud radar to the cirrus optical depth (or ice water content).

Such preliminary analysis is presented in section 3.4 where the Himawari-8 COD retrievals are compared to the radar reflectivity measurements, but these have a few issues. First, CODs up to 25 are detected, how do these classify as cirrus? Second, the Himawari-8 product used here is based on visible/near-IR channels and it is well known that these are not the retrieval of choices for thin cirrus studies, where thermal infrared channels should be preferred. There seems to be Himawari-8 retrievals of ice cloud properties that use such split-window techniques (e.g. Iwabuchi et al, JMSJ, 2018) – these would be much more robust and meaningful. Additionally to the analyses currently shown in section 3.4 the authors should show distributions of Himawari-8 COD and CTH retrievals when the radar detects a cirrus but also when it doesn't, to estimate the amount of missed cirrus and biases in their CTH due to the radar sensitivity. Another possibility would be to use lidar measurements from the same region to detect the cloud occurrence, if any is available to the authors.

Referee#2 made the same comments about the detection ability of KPDR. We use four ways to confirm the detection capability of KPDR, that is, for most cirrus clouds in nature, KPDR can detect them. The confirmation process have been presented in detail in our replies to the comments (A) of reviewer#1. As the content is long, they will not be repeated here. We sincerely ask reviewer to refer to our answers to this comment (from last two paragraphs on Page 2 to Page 5).

We use Himawari-8 COD data because there are more available optical thickness data for analysis when compared with other available COD datasets. In the article, COD data is used only when Himawari-8 determines as cirrus and radar determines as ice cloud. We have no specific analysis for the data with COD greater than 25, and just regard it as the retrieval uncertainty because the COD retrieval of cirrus clouds is inherently challenging. But overall, these abnormal CODs are in the minority, and the final statistics based on large number of data are meaningful.

Reviewer suggested that clouds that AHI measured as cirrus but radar didn't could be used to investigate the radar detection capability. We think this work is a little risky because the field of view of AHI is larger than that of radar. It is very likely that ice clouds appear in the field of view of AHI but not in the field of view of radar. Therefore, it is difficult to explain the contrast results and verify the detection capability. If reviewer agrees, we will not make such an analysis.

3. On a more positive note, the two first comments do not impact the validity of some of the findings such as the seasonal variations (since the same detection threshold applies), or the origin analysis that are based on the reflectivity. There are very interesting finding worth being published and I'd be very much willing to discuss them further after proper clarification of the two aforementioned points in the revised manuscript.

Thanks a lot. We have made revisions in the manuscript carefully according to reviewers' comments. We hope reviewers could accept them.

Specific comments:

Considering the importance and impact that revising the first two points might have on the paper, I have for now limited specific comments to a couple of important points directly related to them.

1. The introduction is very general, it would be more interesting to discuss a bit more the state-of-the- art in radar-only cirrus analyses. How did they tackle the instrumental sensitivity issues to provide usable climatologies?

According to reviewer's suggestion, we add more introductions about this portion in the revised manuscript. For example, "Radar can perform long continuous observations and has high temporal resolution, which is more advantageous than the aircraft in understanding the characteristics of cloud daily changes, the formation and development of clouds. Regular calibration of radar signals can ensure the stability of radar and support long-term data for cloud climatology research."

2. Regarding the occurrence analyses (section 3.2), could you please comment on cirrus detection in the presence of underlying liquid clouds or precipitation? Would that be a limiting factor to the statistics presented here?

If the height gap between two cloudy bins is >150 m (about five radar bins), then the cloud profile is regarded as layered. For a cloud cluster, if more than 80% profiles having single layer, the cloud cluster is regarded as single layer. Please see the left figure below where clouds have two layers. The right figure presents the monthly occurrence frequency of all clouds, all ice clouds and one-layer ice clouds. The one-layer ice clouds dominate over Beijing. Multi-layer cloud is not a limiting factor.

[Figure]

3. In section 3.4 and after, a "mean reflectivity" is used. Could you please define the meaning of this quantity? Is it vertically averaged?

The mean reflectivity is the averaged reflectivity of all cloudy radar bins, not only vertically but also horizontally (namely, time-averaged). We revised the sentence (line 215) to make the meaning more clear.

---

## Referee Report (RR1)

Review #2 of 'Properties of mid-latitude cirrus cloud from surface Ka-band radar observations during 2014-2017' by Huo et al.

I thank the authors for taking the time to respond to my comments. I am pleased with many of the responses and recognized that the manuscript has now gained in quality after this first round of review. However, I still feel that some extra work remains necessary to clarify the goal and result of this study before publication, as detailed below.
* * *
General comments:

1. The definition of "cirrus"and "ice clouds" has been updated and the paper results are consequently easier to interpret. However, the authors still completely disregard the possibility of having mixed-phase clouds. Any ice cloud with a temperature higher than -38C has a high chance (depending on temperature) to include supercooled water. This, as well as the consequences on radar measurements and subsequent climatologies, should be discussed in the paper. The only time they are mentioned is in section 5, where the authors do acknowledge that a significant amount of ice clouds can have an liquid-origin, i.e. originate partly from the supercooled state of the mixed-phase layer. This proves that they also shouldn't be ignored from the previous analyses. Including these clouds in "ice clouds" is not necessarily a problem, as it is often done from remote sensing, but they should still be discussed and their impact on radar measurements detailed.

2. The authors now furhter discuss the sensitivity of the radar in section 2.2 but I am still not completely pleased. The reader needs more quantitative estimates of cloud types that are discussed here or, more importantly, those who are not represented in this study. Please include an estimate of the IWC and OD thresholds, "This KPDR has strong detection capability for ice clouds" (p. 3 l. 68) is not sufficient. I think that the frequency of occurence shown in Fig. 2 (about 4% or less for cirrus) demonstrate that thin clouds are not well detected, and it would be useful to know the detection limit. This threshold should be stated in the abstact as well.

3. It is still difficult to consider that the observations presented in this study are in general terms representative of all "mid-latitude ice clouds". The authors acknowledge this several times within the text, and discuss e.g. in Section 5 specifically of formation mechanism "in Beijing". I would therefore encourage (again) to change the title to "Properties of ice clouds over Beijing from surface Ka-band radar observations during 2014–2017".
* * *
Specific comments:

1. The authors have changed "cirrus" to "ice clouds" but in several encounters it makes no sense (e.g. l. 20, l. 27, l. 78). Please correct.

2. Fig. 2 shows diurnal variations of ice cloud occurence, but I'm not so sure to see the highest oc-curences mentioned p. 6 l. 154. Are they statistically significant? It would be best to include at least standard deviations. Also, how are the ice cloud detections influence by precipitation, which might also occur at specific time of the day?

3. p. 7 l.1 59: What is exactly meant by "extinction process"? If the decrease is indeed robust (see previous point) then the authors should propose some hypothesis at least.

4. In Fig. 5 and Fig. 6, is there a real added value to include the Ze scatterplots rather than the mean and standard deviations? I would suggest to include only the plots (e) and (f) of both figures, together.

5. Same comment concerning Figures 8 to 11, comparing them is really difficult. Why not have only 2 figures that show i) only the overall distribution for all seasons and temperature bins and ii) another similarly one with the PDFs subsetted by thresholds?

6. Section 5: please justify the use for a -34 dBZ threhsold, why this exact value to separate in situ and liquid-origin cirrus?

7. p. 18 l. 365: "superficial" really doesn't sound good, "preliminary" perhaps?

---

## Referee Report (RR2)

Second Review of

**Properties of mid-latitude cirrus cloud
from surface Ka-band radar observations during 2014-2017**

by Juan Huo et al.

For the second review, I start a new discussuion thread, because it will become too confusing to  fill in comments and thoughts again in a different color in the author's response document that went back and forth already several times.

The authors have worked on the main two aspects of the first review, but to my opinion not in a completely satisfactory way. That means  the paper still needs some improvement before it is mature enough to be published.  I will outline this in the  comments below concerning  the two previously mentioned   points   and some new comments that appear with the revised version of the paper.  The comments are sorted in order of appearance in the manuscript. Note that text copied from the manuscript is in italics and quotes.

**1)**  Line 22: '*Cirrus clouds, composed of ice crystals, is ice cloud.*'

First, this sentence is gramatically not correct and second, it is unclear for what reason the sentence should is placed here. Generally, the new text needs some language polishing.

**2)** *Lines 26 – 30: 'Ice clouds exert potential warming effects on the Earth–atmosphere energy system. Studies show that the occurrence frequency of the cirrus clouds, part of ice clouds, exhibits latitudinal variability ranging from 50% in the equatorial regions of Africa to 7% in the polar regions (Hahn and Warren 2007; Sassen et al. 2008, 2009; Stubenrauch et al, 2006).* → see **\*\*** below
*Ice clouds are an important component of the planetary radiation budget in terms of magnitude; plus, they influence hydrological and climate sensitivities and affect surface climate (Lawson et al. 2019; Yang et al. 2015).*'

 **\*\***  Since you also deal with mixed phase clouds, you need to say also something about them at this point.

**3)** Line 45:  please specify the temporal resolution.

**4)** End of Section 2.1 Ka-band radar:   '*It should be noted that KPDR is more sensitive to larger particles in the cloud particle size distribution since the reflectivity is proportional to the D6 (D is particle size).*'

You need to write a sentence here that thin ice clouds containg mostly small ice crystals are not detected!

**5)** Line 78:  '*Cirrus is ice cloud*'   This is no sentence  → see comment 1)

**6)** Section 2.2  Ice cloud identification:  This section is mostly the same as before, describing the the ice cloud observations as they were all cirrus clouds. This  needs to be rewritten  to

place it in the context of ice cloud observation at temperatures < -10 C and then define that those at < -38 C are cirrus.

**7)** Line 169-170: ‚*Both the maximum CTH (13.35 km) ...*'

In line 167 the maximum was defined as 12.9 km.

Does the followign values need to be corrected ?
, *… and the highest mean CTH (10.16 km) are found in summer, whereas winter has the minimum CTH (11.25 km) and lowest mean CTH (7.66 km).*'

**8)** Table 1: all values have slightly changed, why is that ?

**9)** Figure 4 (and respective text):

The maximum cirrus optical depth is reported to be between 1-3, e.g. Sassen et al., 2008; Kienast-Sjögren et al., 2016. In your Figure 4, COD up to 20 is seen, pointing to the glaciated mixed-phase clouds that you have detected. You write in lines 224-225:
‚T*he proportions of CODs lower than 3 in spring, summer, autumn and winter are 46%, 36%, 49% and 52%, respectively.*'
If this portion of ice clouds is at the lower temperatures, then these are probably in-siu-origin cirrus clouds. This could be mentioned in the text.

**10)** Lines 263-264: ‚*At temperatures higher than −38 C, ice clouds can form heteroge-neously or homogeneously (Kanji et al. 2017).*'

This is not correct. At temperatures > −38 C, ice form solely heterogeneously from liquid cloud drops. In case liquid cloud drops are supercooled down to = -38 C, they freeze homogeneously at this temperature. I'm sure that this is correctly described in Kanji et al. (2017).

**11)** Line 291: ‚*... until they are lifted to the ice formation temperature region.*'

… Until they are lifted to temperatures < -38 C.

**12)** Line 300 (and Figures 8-11):

First I want to mention that the new plots are really very intersting!

But:

‚*… central temperature of −65C, −60 C, −55 C, −50 C, −45 C and −40 C...*'

In Figure 6, the lowest detected temperature is -50 C, and in the previous version of the paper, Figure 8 and 9, the lowest center temperature was - 45 C. Where does the new data below -45 C come from ?

(Side comment: I do not understand that you argue in the author's response that ‚*-65 C temperature occupy very small percent of the temperature range*' (your Figure RC6) to explain that there are no cirrus clouds at temperatures < -50 C in your observations – and now such cirrus are presented.

One major comment in my first reviews was that cirrus < -50 C are not present in your data set. But, as the coldest point at mid- latitudes is around – 65 C, absence of these cold, thin cirrus means that they are not detected. We discussed that back and forth and now such cirrus appear … )

– Anyhow, the database is not consistent now, the data at temperatures < -50 C should be added to the observations shown in Figures 5 and 6 for warmer temperatures. Are they included in the analysis shown in Figures 1 – 3 and Table 1? If not, this should also be done.

– Also, in the panels of Figures 5 – 9 the number of data points (or hours of observations) should be noted to give an imprsssion on the statistical significance of the observations.####ol

**13)** Figures 8-11: In the Figure captions is would be good to note in addition that the dashed lines should correspond to in-situ-origin and the dotted lines to liquid-origin.

**14)** Figures 8-11: The PDF's at -45 C and -40 C look very different (much smoother, no modes) than those shown in Figure 8 of the previous version of the manuscript – why is that ?

**15)** Lines 308 ff: The results are very intersting, but mainly only described in the text. It would make the paper scientifically more sound if some ^explanations for the discovered features could be offered. Here are two examples to demonstrate what I mean, but there are more places.

- '*There is no cirrus cloud detected in summer and autumn below −65 C.*'

- In Figures 9 and 10, there are no cirrus clouds below -60 C !
- Could this be because the troposphere is higher in summer and autumn because of higher sun intensity, so the highest and coldest cirrus are not detected?
- The higher sun intensity in summer, causing more convective active, would also explain a higher percentage of liquid-origin cirrus.

- '*Above −55 C, the peak frequency center in winter locates at smaller reflective value than that in summer.*'
What does that mean physically for the cirrus ?

All in all, the results are sound taking into account the underlying processes (which is great), so it would improve the paper to discuss that.

---

## Referee Report (RR3)

Third Review of

**Properties of ice clouds over Beijing**
**from surface Ka-band radar observations during 2014–2017**

by Juan Huo et al.

**General:** The paper has again improved, but my assessment remains that the data set is very interesting, the scientific evaluation and the writing style could still be upgraded. Nevertheless, I'll now recommend the paper for publication with 'minor revisions'. However, I would like to recommend the authors to please submit more mature papers in future to avoid such lengthy review processes with several revision cycles.

**1)** Lines 19-21: ,*Our analysis indicates that most cirrus clouds are of the in situ-origin type in winter and autumn; the in situ-origin type cirrus clouds are more common than liquid-origin cirrus clouds in spring, while summer features more liquid-origin cirrus clouds.'*

→

,Our analysis indicates that in spring, in situ-origin cirrus are more common than liquid-origin cirrus, while in summer liquid-origin cirrus are more frequent; in autumn and winter, most cirrus clouds are of in situ-origin.'

**2)** Line 27: ,*Cirrus clouds are dominated by ice crystals. Studies show that the occurrence frequency of the cirrus clouds exhibits latitudinal variability ...'*

→

,Cirrus clouds consist solely of ice crystals. Their occurrence frequency exhibits latitudinal variability ...'

**3)** Lines 71-79: ,*For comparison, the 94 GHz cloud profiling radar (CPR) on CloudSat has a sensitivity of approximately −30 dBZ.*  This sentence is neither needed nor correct.
*Calculations or measurements of radar reflectivity in previous studies reveal that the reflectivity of ice clouds over mid-latitude regions are mostly larger than −30 dBZ (Deng et al., 2010; Pokharel and Vali, 2011; Matrosov and Heymsfield, 2017). Therefore, KPDR is capable of detecting most ice clouds over Beijing. However, the Ka- band radar is more sensitive to larger particles in a cloud target since the reflectivity is proportional to the D6 (D is particle size). For the CPR, thin ice clouds with ice water content (IWC) lower than approximately 0.4 mg/m3 are invisible (Wu et al., 2009). It is possible that KPDR  miss some thin ice clouds when they  consist of  small ice crystals (i.e., D less than 20 μm) or the IWC is smaller than 0.4 mg/m3.'*

**4)** Line 88: ,*Ice cloud  are composed of various types of ice crystals and  are usually thin.'*

**5)** point 12) of the last review:

A logarithmic color code is usually used for a wide range of values to be displayed. Below is an example from the Internet - please redo Figures 5 and 6 in this way:

[Figure]

**6)** point 15) of the last review, new lines 334-339:

*‚In the four seasons, the 'n' at −45°C is the biggest among all temperatures, indicating that cirrus cloud appears more frequently at −45°C than at other temperatures. This is the result of the combination of temperature, water vapor and upward movement. For example, the lower the temperature, the more conducive it is to the formation of ice particles* (this is not correct). *On the other hand, limited upward movements determine the maximum height (lowest temperature) where water vapor or cloud particles can reach* (this is not correct). *Spring has the biggest 'n' at each temperature, indicating that cirrus clouds in spring are the most frequent, which has also been shown in Fig. 2.'*

The finding that cirrus  clouds appear most frequent at −45°C is consistent with the findings of Krämer et al. (2020) from 10 years of satellite observations (what is great!). The reason is that at these altitudes both in situ origin  and liquid origin cirrus appear, whereas at colder temperatures only in situ origin cirrus exist. Please change the explanation accordingly.

Krämer, M., Rolf, C., Spelten, N., Afchine, A., Fahey, D., Jensen, E., Khaykin, S., Kuhn, T., Lawson, P., Lykov, A., Pan, L. L., Riese, M., Rollins, A., Stroh, F., Thornberry, T., Wolf, V., Woods, S., Spichtinger, P., Quaas, J., and Sourdeval, O.: A Microphysics Guide to Cirrus – Part II: Climatologies of Clouds and Humidity from Observations, Atmos. Chem. Phys. Discuss., https://doi.org/10.5194/acp-2020-40, in review, 2020.

---

## Author Response (AR2)

**Responses to Reviewer**

We deeply thank reviewer for the helpful comments and suggestions which present us important guidance to the improvements our researches and the manuscript. We have made corresponding corrections and added more descriptions in the revised manuscript according to the comments. Our responses below to the comments are shown in blue. Revisions in the manuscript are marked with same color. The language in the manuscript has been edited by a professional and native-English editor. In the revised manuscript, we also made some corrections on Figure 1-11 to get a well-formed and clear figure. Formats of some references were also corrected. We hope our responses and modifications can meet with your approval.

**Review #1** of Properties of mid-latitude cirrus cloud from surface Ka-band radar observations durin2014-2017 by Huo et al.

I thank the authors for taking the time to respond to my comments. I am pleased with many of the responses and recognized that the manuscript has now gained in quality after this first round overview. However, I still feel that some extra work remains necessary to clarify the goal and result of this study before publication, as detailed below.

**General comments:**

1. The definition of "cirrus" and "ice clouds" has been updated and the paper results are consequently easier to interpret. However, the authors still completely disregard the possibility of having mixed phase clouds. Any ice cloud with a temperature higher than -38C has a high chance (depending on temperature) to include supercooled water. This, as well as the consequences on radar measurements and subsequent climatologies, should be discussed in the paper. The only time they are mentioned is in section 5, where the authors do acknowledge that a significant amount of ice clouds can have a liquid-origin, i.e. originate partly from the supercooled state of the mixed-phase layer. This proves that they also shouldn't be ignored from the previous analyses. Including these clouds in "ice clouds" is not necessarily a problem, as it is often done from remote sensing, but they should still be discussed and their impact on radar measurements detailed.

Ice clouds mostly contain ice crystals. We agree that supercooled water should exist in ice clouds. According to reviewer's advice, we added some discussions about the supercooled water in section 2.2 (last sentence).

However, what's the proportion of supercooled water in ice cloud over Beijing? Can it be neglected? We searched in publications but did not find useful answer. Could reviewer please give us some references? And, the identification method of supercooled water from KPDR data need to be developed at present. We hope review could allow us to make such investigations in future.

2. The authors now further discuss the sensitivity of the radar in section 2.2 but I am still not completely pleased. The reader needs more quantitative estimates of cloud types that are

discussed here or, more importantly, those who are not represented in this study. Please include an estimate of the IWC and OD thresholds, "This KPDR has strong detection capability for ice clouds "(P3. L. 68) is not sufficient. I think that the frequency of occurrence shown in Fig. 2 (about 4% or less for cirrus) demonstrate that thin clouds are not well detected, and it would be useful to know the detection limit. This threshold should be stated in the abstract as well.

Did reviewer mention to section 2.1? From the content, we think it is section 2.1 not 2.2. According to reviewer's suggestion, we added more explanations and discussions about the capability of Ka radar in section 2.1 (line71-79).

As shown in Equation (4) and (5) in the manuscript, IWC and OD is related with the number density, particle size, particle distribution function, and particle shape. Radar reflectivity factor is also related with these parameters. The statistical relationship between IWC and reflectivity is presented in Equation (5) with various A and B for different cloud types at different temperature and places. Thus, reflectivity is also connected with the properties of cloud target. Wu et al. (2019) reported that for the 94 GHz cloud profiling radar (CPR) on CloudSat, thin ice clouds with IWC lower than approximately 0.4 mg/m$^3$ are invisible by comparison with MSL. CPR has a sensitivity of approximately −30 dBZ. This work gave us a reference.

Synchronous measurements by lidar are useful to validate the detection capability of Ka radar in cirrus clouds. There has been a lidar and a Ka radar making daily measurements at our Tibet observatory since last year. We will make such investigations when data are met.

3. It is still difficult to consider that the observations presented in this study are in general terms representative of all mid-latitude ice clouds". The authors acknowledge this several times within the text, and discuss e.g. in Section 5 specifically of formation mechanism "in Beiing". I would therefore encourage (again) to change the title to "Properties of ice clouds over Beijing from surface Ka-bandradar observations during 2014-2017".

We revised the title according to reviewer's advice.

**Specific comments:**

1. The authors have changed "cirrus " to "ice clouds " but in several encounters it makes no sense(e.g 1. 20, 1. 27, 1. 78). Please correct.

According to the comments from both reviewers, more revisions were made in the Introduction (e.g. l. 22-23, l.27-32).

2. Fig 2 shows diurnal variations of ice cloud occurrence. but I'm not so sure to see the highest occurrences mentioned p. 6 l. 154. Are they statistically significant? It would be best to include at least standard deviations. Also, how are the ice cloud detections influence by precipitation, which might also occur at specific time of the day?

They are not statistically significant. These occurrence frequency were calculated from whole dataset using the equation (1) and (2), not the average values of the four years. So, no standard deviation is calculated. According to our analysis (Huo et al 2020, in Chinese), the average precipitation frequency from all clouds is about 5% during 2014-2017. The

precipitation from ice clouds is lower than 1%. Thus, the influence from precipitation is not considered in this paper since it makes little contribution.

3. p. 7 1.159: What is exactly meant by "extinction process"? If the decrease is indeed robust (see previous point) then the authors should propose some hypothesis at least.

Sorry for that. It was revised as decay process. Some hypothesis was added (please see l.170-171).

4. In Fig. 5 and Fig. 6, is there a real added value to include the Ze scatter plots rather than the mean and standard deviations? I would suggest to include only the plots (e) and (f) of both figures, together.

Figure 5 and Figure 6 were obtained from large number of data which cannot be presented wholly within a scatter plot. To keep the symmetry, we plot the mean and standard deviation in separate panel. We hope reviewer could accept them.

5. Same comment concerning Figures 8 to 11, comparing them is really difficult. Why not have only 2figures that show i)only the overall distribution for all seasons and temperature bins and ii)another similarly one with the PDFS subsetted by thresholds?

We have thought to show the frequency of four seasons in one figure. But the figure became messy when the origin-type lines were added since those lines were different in four seasons. In addition, the maximum numbers used to calculate the normalization frequency are various among seasons and temperatures. Considering these reasons, we showed them in four figures in order to present a clear result. We hope reviewer could accept them.

6. Section 5: please justify the use for a-34 dBZ threhsold, why this exact value to separate in situ and liquid-origin cirrus?

This work is based on the hypothesis that cirrus clouds over Beijing originate by two types. Then, we tried every threshold between -32 ~ -38 dBZ and found that cirrus clouds in spring and summer were separated clearly into two groups by the threshold of -34 dBZ and the two groups demonstrated different PDFs. Then threshold of -34 dBZ was selected.

7. P. 18 365: superficial really doesn't sound good, "preliminary" perhaps?

Sorry for that, it was revised according to reviewer's advice.

**Review #2** Properties of mid-latitude cirrus cloud from surface Ka-band radar observations during 2014-2017 by Juan Huo et al.

For the second review, I start a new discussuion thread, because it will become too confusing to fill in comments and thoughts again in a different color in the author's response document that went back and forth already several times.

The authors have worked on the main two aspects of the first review, but to my opinion not in a completely satisfactory way. That means the paper still needs some improvement before it is mature enough to be published. I will outline this in the comments below concerning the two previously mentioned points and some new comments that appear with the revised version of the paper. The comments are sorted in order of appearance in the manuscript. Note that text copied from the manuscript is in italics and quotes.

1) Line 22: '*Cirrus clouds, composed of ice crystals, is ice cloud.*'
First, this sentence is gramatically not correct and second, it is unclear for what reason the sentence should is placed here. Generally, the new text needs some language polishing.
   This sentence was revised as "Cirrus clouds are dominated by ice crystals". This revised manuscript has been polished by a native English editor.

2) Lines 26 – 30: *‚Ice clouds exert potential warming effects on the Earth–atmosphere energy system. Studies show that the occurrence frequency of the cirrus clouds, part of ice clouds, exhibits latitudinal variability ranging from 50% in the equatorial regions of Africa to 7% in the polar regions (Hahn and Warren 2007; Sassen et al. 2008, 2009; Stubenrauch et al, 2006).* → see ✱✱ below
*Ice clouds are an important component of the planetary radiation budget in terms of magnitude; plus, they influence hydrological and climate sensitivities and affect surface climate (Lawson et al. 2019; Yang et al. 2015).*'
✱✱ Since you also deal with mixed phase clouds, you need to say also something about them at this point.
   Yes. According to reviewer's advice, we added some descriptions about the ice clouds in the Introduction (please see Line 27-31).

3) Line 45: please specify the temporal resolution.
   Yes. According to reviewer's advice, we added an example since temporal resolution changes between radars.

4) End of Section 2.1 Ka-band radar: *‚It should be noted that KPDR is more sensitive to larger particles in the cloud particle size distribution since the reflectivity is proportional to the D6 (D is particle size).*'
You need to write a sentence here that thin ice clouds containg mostly small ice crystals are not detected!
   According to the advices from both reviewers, we added more descriptions about the capability in this section (please see Line 70-79).

5) Line 78: *‚Cirrus is ice cloud*' This is no sentence → see comment 1)
   It was deleted.

6) Section 2.2 Ice cloud identification: This section is mostly the same as before, describing the ice cloud observations as they were all cirrus clouds. This needs to be rewritten to place it in the context of ice cloud observation at temperatures < -10 C and then define that those at < -38 C are cirrus.

According to reviewer's advice, we revised this section. The structure is rearranged and some descriptions are added. We hope they could satisfy reviewer.

7) Line 169-170: '*Both the maximum CTH (13.35 km) ...*' In line 167 the maximum was defined as 12.9 km.
Does the followign values need to be corrected?
'*... and the highest mean CTH (10.16 km) are found in summer, whereas winter has the minimum CTH (11.25 km) and lowest mean CTH (7.66 km).*'

Very sorry for our carelessness. The old ones are forgotten to revise. They have been replaced with new statistical results. All statistical numbers in this section (section 3) have been examined and corrected in the revised manuscript.

8) Table 1: all values have slightly changed, why is that ?

In the revised manuscript, in order to select the ice clouds with high confidence, we used a new threshold ("mean cloud-top temperature less than −40°C and the maximum cloud-base temperature less than -10°C") for ice cloud identification, not the old "the cloud-top temperature should be less than −30°C and the cloud-base temperature should be less than 0°C". With the new criteria, some cloud clusters are excluded. About half of them are from summer. Therefore, statistical results based on these new ice clouds selected by new criteria changed.

9) Figure 4 (and respective text):
The maximum cirrus optical depth is reported to be between 1-3, e.g. Sassen et al., 2008; Kienast-Sjögren et al., 2016. In your Figure 4, COD up to 20 is seen, pointing to the glaciated mixed-phase clouds that you have detected. You write in lines 224-225:
'*The proportions of CODs lower than 3 in spring, summer, autumn and winter are 46%, 36%, 49% and 52%, respectively.*'
If this portion of ice clouds is at the lower temperatures, then these are probably in-siu-origin cirrus clouds. This could be mentioned in the text.

According to reviewer's advice, we illustrate the ice clouds with cloud base temperature lower than -38 in new Fig.4. The related text were also revised (please see Line 236-243).

10) Lines 263-264: '*At temperatures higher than −38 C, ice clouds can form heteroge- neously or homogeneously (Kanji et al. 2017).*'
This is not correct. At temperatures > −38 C, ice form solely heterogeneously from liquid cloud drops. In case liquid cloud drops are supercooled down to = -38 C, they freeze homogeneously at this temperature. I'm sure that this is correctly described in Kanji et al. (2017).

Sorry for that. It has been revised as "At temperatures higher than −38°C, primary ice clouds form only when aided by ice nucleating particles (Kanji et al., 2017).".

11) Line 291: '*... until they are lifted to the ice formation temperature region.*'
… Until they are lifted to temperatures < -38 C.

It was revised as reviewer's advice.

12) Line 300 (and Figures 8-11):
First I want to mention that the new plots are really very intersting! But:
'*... central temperature of −65C, −60 C, −55 C, −50 C, −45 C and −40 C...*'
In Figure 6, the lowest detected temperature is -50 C, and in the previous version of the paper, Figure 8 and 9, the lowest center temperature was - 45 C. Where does the new data below -45 C come from?
(Side comment: I do not understand that you argue in the author's response that '*-65 C temperature occupy very small percent of the temperature range*' (your Figure RC6) to explain that there are no cirrus clouds at temperatures < -50 C in your observations – and now such

cirrus are presented. One major comment in my first reviews was that cirrus < -50 C are not present in your data set. But, as the coldest point at mid- latitudes is around – 65 C, absence of these cold, thin cirrus means that they are not detected. We discussed that back and forth and now such cirrus appear … )

Due to insufficient descriptions and inappropriate expressions, we are sorry to bring reviewer so many doubts.

First of all, we need to state that all the analysis results, including every table and figure in this revised article, come from the same dataset.

Some ice clouds with temperatures below $-50^\circ$ C are shown in Figures 8-11, but they are not shown in Figure 6, which is related to the plotting way of Figure 6 and the data itself. The frequency is represented by the statistical counts calculated at 0.25 dBZ and 1 °C interval. The variation range of these counts is too wide (from 300 to 500,000). In Figure 6, the statistical counts was divided by 5000 firstly and then displayed with color by the color map for aesthetic reasons. Our naked eyes couldn't d distinguish those very low values because they are two small relative to 500,000 although they have been displayed in Figure 6. We tried other plotting ways. For example, the figure below show the original counts, not divided by 5000 with different color map). Similarly, these low counts are also hard to distinguish from the background. Thus, the ice clouds below $-50\,^\circ$ C shown in Figures 8 to 11 are real, but compared to the ice clouds above -50 $^\circ$ C, the counts is too small to be shown clearly in Figure 6. Since purpose of this section is to investigate the dominated temperatures and reflectivity, current Fig.6 is appropriate to show the key results. In order to reduce the confusion of readers, we added relevant explanations in the text for Figure 6. Also, quantities represented by the color bar are added. In addition, the maximum counts in each panel in Figures 8~11 are also provided for comparison. It should be noted that the counts in Figures 8~11 are calculated within the range of $T \pm 1\,^\circ$ C interval while in Figure 6 it is calculated within $T \pm 0.5^\circ$ C.

[Figure]

– Anyhow, the database is not consistent now, the data at temperatures < -50 C should be added to the observations shown in Figures 5 and 6 for warmer temperatures. Are they included in the analysis shown in Figures 1 – 3 and Table 1? If not, this should also be done.

As those answers above, each table and figure in the revised manuscript is obtained from same database.

– Also, in the panels of Figures 5 – 9 the number of data points (or hours of observations) should be noted to give an impresssion on the statistical significance of the observations.

According to reviewer's advice, hours of the observations are added in this section (please see line 85). That is, there are more than 28,000 hours of measurements during the period 2014-

2017.

13) Figures 8-11: In the Figure captions is would be good to note in addition that the dashed lines should correspond to in-situ-origin and the dotted lines to liquid-origin.

According to reviewer's advice, they were added in the captions.

14) Figures 8-11: The PDF's at -45 C and -40 C look very different (much smoother, no modes) than those shown in Figure 8 of the previous version of the manuscript – why is that ?

It is because that data used for analysis changed.

(1)The previous criteria for identifying ice cloud is the temperature of the cloud top should be less than −30°C and the temperature at the maximum Ze layer and at the cloud base should be less than 0°C. The new criteria in the revised paper is that mean cloud-top temperature less than −40°C and the maximum cloud-base temperature less than -10°C. There are about 300 cloud clusters are excluded. Some of them are clouds with thick depth, for example > 5 km. About half of them are from summer.

(2) Previous Figure 8 shows the PDF of all ice clouds (the "−30°C; 0°C threshold"). The new Figures 8-11 show the PDF of all cirrus clouds (ice clouds with cloud-base temperature lower than -38°C).

15) Lines 308 ff: The results are very intersting, but mainly only described in the text. It would make the paper scientifically more sound if some ^explanations for the discovered features could be offered. Here are two examples to demonstrate what I mean, but there are more places.

- '*There is no cirrus cloud detected in summer and autumn below −65 C.*'

• In Figures 9 and 10, there are no cirrus clouds below -60 C !
• Could this be because the troposphere is higher in summer and autumn because of higher sun intensity, so the highest and coldest cirrus are not detected?
• The higher sun intensity in summer, causing more convective active,would also explain a higher percentage of liquid-origin cirrus.

- '*Above −55 C, the peak frequency center in winter locates at smaller reflective value than that in summer.*'
What does that mean physically for the cirrus ?

All in all, the results are sound taking into account the underlying processes (which is great), so it would improve the paper to discuss that.

According to reviewer's advice, we added more explanations and discussions in this section (e.g. Line 329-333, Line 334-339). Please see the new section 5. Thank reviewer very much for these comments and suggestions.

---

## Author Response (AR3)

**Response to editor and reviewer's comments**

Thank editor and reviewers very much again. Our responses to the comments are presented in blue. Minor revisions associated with the comments are marked with same color in the manuscript. In addition, we revised the formats of some references according to the ACP's word reference template. Since there are too many minor modifications in the references, they are not marked.

Comments to the Author:

Based on the referees suggestions, I accept the paper now after minor revisions. Please consider the last points the referees raised, then the paper can be published in ACP.

I have one further point from my side. I saw another paper where data analyzes based on the same data set are presented:

> J Huo, C Han, M Duan, X Wu, Y Bi, Y Tian: Particle reflectivity and movements in cirrus clouds over Beijing from four years of Ka radar measurements, Atmospheric Research, https://doi.org/10.1016/j.atmosres.2020.105211, available online 22 August, in Press, Journal Pre-proof, 2020.

In this paper it is stated that it is an extension of the ACPD manuscript. So I think the Atmospheric Research paper should be referenced in the ACPD paper and suggest to discuss and compare the ACPD results to that of the Atmospheric Research paper, maybe in a short extra section at the end of the manuscript.

Thank the referee. We added some sentences about our work accepted by the Atmospheric Research at the end of the manuscript.

---

## Author Response (AR4)

**Response to editor and reviewer's comments**

Sorry for missing revisions and responses to referees' comments. Our responses to the comments are presented below in dark green. Minor revisions associated with the comments are marked with same color in the manuscript.

Comments to the Author from referee #1:

I thank the authors for their effort in revising the manuscript. I am particularly pleased that discussions of the results are now more precise, backed with references and straight to the point. The title now also fits the paper much better.

One point that could still be argued about is that of mixed-phase clouds, but I'll agree with the authors in these are difficult to quantify and deal with from a remote sensing point of view. Mentioning them (as done in section 2.2) should be sufficient for this paper. However, I would like to see that discussion highlighted again in the conclusion so that the current limitation due to ignoring mixed-phase clouds or supercooled layers in current results is properly reminded to the reader. Provided the abovementioned addition to the conclusion in a revised manuscript, I am happy to suggest for accepting the paper to ACP.

Thank referee. A sentence about current limitation in the results has been added in last paragraph with dark green.

Comments to the Author from referee #2:

**General:** The paper has again improved, but my assessment remains that the data set is very interesting, the scientific evaluation and the writing style could still be upgraded. Nevertheless, I'll now recommend the paper for publication with 'minor revisions'. However, I would like to recommend the authors to please submit more mature papers in future to avoid such lengthy review processes with several revision cycles.

Thank the editors and reviewers very much for the patience. We are very grateful to you for your comments on this article, based on which research results and quality of the article are improved. In addition, we have also benefited a lot from them. Our knowledge of the radar detection ability of cirrus and cirrus clouds related microphysics gets more accurate. Thanks again.

1) Lines 19-21: ‚Our analysis indicates that most cirrus clouds are of the in situ-origin type in winter and autumn; the in situ-origin type cirrus clouds are more common than liquid-origin cirrus clouds in spring, while summer features more liquid-origin cirrus clouds.‘

→

, Our analysis indicates that in spring, in situ-origin cirrus are more common than liquid origin cirrus, while in summer liquid-origin cirrus are more frequent; in autumn and winter, most cirrus clouds are of in situ-origin.‘

Revisions have been made according to referee's suggestion.

2) Line 27: ‚Cirrus clouds are dominated by ice crystals. Studies show that the occurrence frequency of the cirrus clouds exhibits latitudinal variability ...‘

→

, Cirrus clouds consist solely of ice crystals. Their occurrence frequency exhibits latitudinal variability ...‘

Revisions have been made according to referee's suggestion.

3) Lines 71-79: ,*For comparison, the 94 GHz cloud profiling radar (CPR) on CloudSat has a sensitivity of approximately −30 dBZ.*  This sentence is neither needed nor correct. *Calculations or measurements of radar reflectivity in previous studies reveal that the reflectivity of ice clouds over mid-latitude regions are mostly larger than −30 dBZ (Deng et al., 2010; Pokharel and Vali, 2011; Matrosov and Heymsfield, 2017). Therefore, KPDR is capable of detecting most ice clouds over Beijing. However, the Ka- band radar is more sensitive to larger particles in a cloud target since the reflectivity is proportional to the D6 (D is particle size). For the CPR, thin ice clouds with ice water content (IWC) lower than approximately 0.4 mg/m3 are invisible (Wu et al., 2009). It is possible that KPDR  miss some thin ice clouds when they  consist of  small ice crystals (i.e., D less than 20 μm in effective diameterm) or the IWC is smaller than 0.4 mg/m3.'*
Revisions have been made according to referee's suggestion.

4) Line 88: ,Ice cloud  are composed of various types of ice crystals and  are usually thin.'
Revisions have been made according to referee's suggestion.

5) point 12) of the last review:
A logarithmic color code is usually used for a wide range of values to be displayed. Below is an example from the Internet - please redo Figures 5 and 6 in this way:

[Figure]

https://www.researchgate.net/figure/Both-figures-showoccurrence-
of-tracking-errors-in-logarithmic-color-code-Theleft_fig2_281982255

We modify Figures 5 and 6 using the log number way.

6) point 15) of the last review, new lines 334-339:
, In the four seasons, the 'n' at −45°C is the biggest among all temperatures, indicating that cirrus cloud appears more frequently at −45°C than at other temperatures. This is the result of the combination of temperature, water vapor and upward movement. For example, the lower the temperature, the more conducive it is to the formation of ice particles (this is not correct). On the other hand, limited upward movements determine the maximum height (lowest temperature) where water vapor or cloud particles can reach (this is not correct). Spring has the biggest 'n' at each

temperature, indicating that cirrus clouds in spring are the most frequent, which has also been shown in Fig. 2.'

The finding that cirrus clouds appear most frequent at −45°C is consistent with the findings of Krämer et al. (2020) from 10 years of satellite observations (what is great!). The reason is that at these altitudes both in situ origin and liquid origin cirrus appear, whereas at colder temperatures only in situ origin cirrus exist. Please change the explanation accordingly.

Krämer, M., Rolf, C., Spelten, N., Afchine, A., Fahey, D., Jensen, E., Khaykin, S., Kuhn, T., Lawson, P., Lykov, A., Pan, L. L., Riese, M., Rollins, A., Stroh, F., Thornberry, T., Wolf, V., Woods, S., Spichtinger, P., Quaas, J., and Sourdeval, O.: A Microphysics Guide to Cirrus – Part II: Climatologies of Clouds and Humidity from Observations, Atmos. Chem. Phys. Discuss., https://doi.org/10.5194/acp-2020-40, in review, 2020.

Yes. The reason suggested by referee revealed the essences. Revisions have been made according to referee's suggestion.

---

## Author Response (AR5)

**Response to editor's comments**

Thank editor very much. Our responses to each comment are presented below in blue. Revisions in the manuscript are marked with same color.

1) please check the new Figures 5 and 6 for the following reason:

when comparing the new with the old Figures, the only difference that I can see is the color code. This is strange since you switched from linear to logarithmic color code. The reason to do that was to make the data points below about -50C (shown in Figure 8) visible, but these points are still not seen. The color at the low temperatures is blue, which means 1 data point in each data interval. This is not in agreement with the data numbers noted in Figure 8 at the lower temperatures.

Initially, these data were not shown in figures 5 and 6, and we took it for granted that the values were too low to be distinguished from the background (dark blue). Sorry for our precipitance. According to editor's suggest, we checked the data and program codes of figures 5 and 6. In the data source, there are data with the values greater than 10 in the low temperature region. After checking the imaging codes, it is found that in the process of pseudo-data with dBZ < - 45 (threshold of ice cloud), some data with relatively low logarithm value are also wrongly assigned with zero, resulting in incomplete display using invalid colormap projection. We are very sorry for that. The revised figures 5 and 6 should be correct.

2) Figure 6, caption: it is still noted that the number of the counts is divided by 5000, I don't think that is the case.
Sorry. "divided by 5000" should be deleted.

3) Point 6) of the last review, new lines 334-335: new text inserted in the paper: 'The reason is that at these altitudes both in situ origin and liquid origin cirrus appear, whereas at colder temperatures only in situ origin cirrus exist.' Since this is one of the findings of Krämer et al. (2020), to my feeling this paper could be cited.
Sorry. This reference has been cited.